# Nonlocal Analysis of the Flexural–Torsional Stability for FG Tapered Thin-Walled Beam-Columns

**DOI:** 10.3390/nano11081936

**Published:** 2021-07-27

**Authors:** Masoumeh Soltani, Farzaneh Atoufi, Foudil Mohri, Rossana Dimitri, Francesco Tornabene

**Affiliations:** 1Department of Civil Engineering, Faculty of Engineering, University of Kashan, Kashan 8731753153, Iran; atoufi1373@gmail.com; 2Université de Lorraine, CNRS, Arts et Métiers ParisTech, LEM3, LabEx DAMAS, F-57000 Metz, France; foudil.mohri@univ-lorraine.fr; 3Department of Innovation Engineering, University of Salento, 73100 Lecce, Italy; francesco.tornabene@unisalento.it

**Keywords:** axially functionally graded materials, differential quadrature method, flexural–torsional buckling, nonlocal elasticity theory, tapered I-beam

## Abstract

This paper addresses the flexural–torsional stability of functionally graded (FG) nonlocal thin-walled beam-columns with a tapered I-section. The material composition is assumed to vary continuously in the longitudinal direction based on a power-law distribution. Possible small-scale effects are included within the formulation according to the Eringen nonlocal elasticity assumptions. The stability equations of the problem and the associated boundary conditions are derived based on the Vlasov thin-walled beam theory and energy method, accounting for the coupled interaction between axial and bending forces. The coupled equilibrium equations are solved numerically by means of the differential quadrature method (DQM) to determine the flexural–torsional buckling loads associated to the selected structural system. A parametric study is performed to check for the influence of some meaningful input parameters, such as the power-law index, the nonlocal parameter, the axial load eccentricity, the mode number and the tapering ratio, on the flexural–torsional buckling load of tapered thin-walled FG nanobeam-columns, whose results could be used as valid benchmarks for further computational validations of similar nanosystems.

## 1. Introduction

Thin-walled beams with open cross-sections (e.g., channel, angle, I- and Tee-sections) carry an extensive variety of potential applications as structural components in various engineering fields (from civil to aeronautical engineering) since they offer high performances with a minimal weight. Moreover, thin-walled beams with varying cross-sections have been of great interest to designers and researches, especially in recent decades. The optimization of weight, the reduction in volume, and the improvement of both strength and stability represent some crucial reasons to increase their use as structural members. Due to the low torsion stiffness, a slender beam with a thin-walled cross-section subjected to an eccentric compressive axial force can buckle in the flexural–torsional mode. Thus, investigations about the stability of tapered thin-walled beams can be very complicated because of the coupled bending and torsional deformations involved, as well as the arbitrary variation in the geometrical properties along the longitudinal direction.

As far as advanced multi-phase composites are concerned, functionally graded materials (FGMs) represent a novel generation of composite materials, based on a smooth and gradual variation in the volume fraction of their constituent phases in any desired direction. Compared to traditional materials and laminated composites, FGMs possess some important advantages, primarily, multifunctionality, a high temperature-withstanding ability, the reduction or total removal of stress concentrations, together with the improved strength and fracture toughness. Due to these favorable features, FGMs can represent ideal materials for the design of smart engineering systems and devices, which has motivated their recent extensive use in many engineering applications and modern industries, such as aerospace, automobile, optics, nuclear, electronic and turbine components.

With the recent development of nanotechnology, nanoscaled structural elements such as nanobeams and nanoplates are being widely used as key components in different modern engineering devices, including sensors, actuators, transistors, probes, and nano-electromechanical systems (NEMS). This requires an appropriate study of the mechanical properties of similar structural systems, with even more complicated natures. The experimental tests demonstrate that classical continuum theories cannot be implemented for the exact analysis of nanostructures, as the size effect can play a significant role in their mechanical behavior. Thus, various higher-order size-dependent continuum theories, such as the modified couple stress theory [1], the surface energy theory [2] and nonlocal elasticity theory [3,4], have been expanded to model small-sized structures. Among these models, the nonlocal elasticity theory, as suggested by Eringen [3], has been widely used in the literature to investigate the stability, deformation and vibrational responses of nanostructural elements, assuming that the stress state at an arbitrary point in a body depends not only on the strain field at that point, but also on the strain fields at all points of the body. At the same time, FGMs have been increasingly applied in small-sized structures due to their superior mechanical properties. In such a context, over the past few years, several investigations have been performed to study the linear and nonlinear mechanical responses of nanosized structures made from homogenous or FGMs. Moreover, a large number of works can be found in the literature focusing on the elastic and/or inelastic static, vibration and instability behavior of beams with a thin-walled cross-section, due to their vast relevance in many engineering configurations. Among the most relevant works on the topic, Kitipornchai and Trahair [5] determined the flexural–torsional critical force of doubly- and/or singly-symmetric I-beams with a geometrical variation under non-uniform torsion. Wekezer [6,7] studied the stability of thin-walled beams with varying open sections based on shell theory strain tensors. Considering the influence of geometric nonlinearity, a finite element technique was suggested by Yang and Yau [8] to assess the buckling behavior of doubly symmetric tapered I-beams. Bradford and Cuk [9] adopted a novel finite element technique to determine the buckling limit state of web-tapered beams with a mono-symmetric I-section. In another study, web-tapered beams with a Tee-section were probed by Baker [10]. A finite element formulation was also applied by Rajasekaran [11,12] to approximate the linear stability resistance of tapered thin-walled beams. Similarly, a simple finite element solution was presented by Gupta et al. [13] and Ronagh et al. [14] to predict the lateral–torsional resistance of tapered I-beams. With the help of the total potential energy and Hamilton’s principle, Chen [15] computed the vibrational properties of thin-walled beams with geometrical variation. An innovative finite element formulation was also proposed by Kim [16] to analyze the lateral–torsional buckling (LTB) and vibration behavior of beams with a tapered I-section under different boundary conditions. The shear deformation effect was also accounted within the formulation of Li [17] for the stability study of beams with a linearly variable cross-section under a compressive axial load. A nonlocal elasticity version was also suggested by Peddieson et al. [18] to elaborate a nonlocal Benoulli/Euler beam model. A semi-inverse approach was then employed by Elishakoff et al. [19] for the vibrational analysis of beams made of axially-inhomogeneous materials, whereas Refs. [20,21,22] represent some further useful contributions to the lateral–torsional stability study of thin-walled beams with doubly- and singly-symmetric I-sections under different boundary conditions. Taking into account small deformations and large displacements, Mohri et al. [23,24] analyzed the nonlinear flexural–torsional behavior of thin-walled beams with arbitrary cross-sections by employing the Galerkin method, while Samanta and Kumar [25] provided a shell finite element solution for the study of the distortional buckling resistance of beams with a singly-symmetric I-section under simply supports.

In the field of nonlocal differential elasticity methodology, Reddy [26] proposed some pioneering analytical solutions for the static, buckling and vibrational analyses of beams by considering different shear deformation theories. Some additional analytical outcomes for cantilever beams with linear tapered section were also presented by Challamel et al. [27]. Wang et al. [28] perused the flexural vibration problem of nano- and microbeams, following the assumptions of the nonlocal elasticity theory of Eringen in conjunction with the Timoshenko beam model. Many further works in the literature have successfully applied the Eringen nonlocal elasticity approach combined with different beam theories and numerical solution methods—see Refs. [29,30,31,32,33,34,35,36,37,38]. Among them, Pradhan and Sarkar [29] studied the deformation, instability and vibrational responses of an Euler–Bernoulli beam with variable geometrical and material properties. Aydogdu [30] derived a generalized nonlocal beam theory for the mechanical analysis of nanosize beams by means of the Eringen elasticity assumptions combined with different beam theories. In the same direction, Civalek and Akgöz [31] studied the free vibrational properties of microtubules, which problem was solved numerically based on the DQM. Danesh et al. [32] determined the equations of motion for the longitudinal vibration of nanorods with tapered cross-sections, and solved them via the DQM. Şimşek and Yurtcu [33] used the Timoshenko beam theory to survey the deformation and buckling capacity of nanobeams with varying materials. McCann et al. [34] studied the lateral buckling resistance of steel beam members under pure bending and with simply-supported ends, in presence of discrete elastic lateral restraints along their axial direction. Following the nonlocal continuum theoretical assumptions, a finite element formulation was suggested by Eltaher et al. [35,36] to assess the size effect on the mechanical response of nanobeams made of FG materials. An Euler–Bernoulli beam model was also proposed by Shahba et al. [37] to compute the critical axial forces and natural frequencies of tapered beams with axially non-homogeneous materials. Within the framework of large torsion, Benyamina et al. [38] developed a nonlinear formulation to analyze the lateral stability and buckling moment of tapered I-section beams under simply–simply supports. Among the different numerical strategies to handle similar problems, Nguyen et al. [39] proposed an approximate methodology to evaluate the critical moment of I-section beams in the presence of discrete torsional bracing. Attard and Kim [40] included the shear deformations to determine the lateral stability equations for isotropic beams with a thin-walled open section. Challamel and Wang [41] employed Bessel functions for an exact computation of the lateral–torsional buckling load of strip cantilever beam members subjected to an arbitrary loading distribution. A modified couple stress theory was differently combined with the first-order shear deformable beam model of Ke et al. [42] to describe the size effect on the dynamic stability of microbeams made of FGMs. A novel finite element solution was proposed by Borbon [43] to study the coupled vibrational responses of beams with non-symmetric thin-walled cross-sections, accounting for the possible influence of loading eccentricities, shear deformation and rotatory inertia, and a further approximate methodology was successfully introduced by Serna et al. [44] to study the elastic flexural buckling of non-uniform columns subjected to arbitrary axial forces. Akgoz and Civalek [45] surveyed the free vibrational problem of axially functionally graded (AFG) non-uniform microbeams based on a Euler–Bernoulli beam model and modified couple stress theory. In order to exhaustively assess the static and dynamic responses of beams made of FG piezoelectric materials, an improved three-noded beam element was formulated by Lezgy-Nazargah et al. [46], whereas a novel beam finite element was developed by Trahair [47] for the lateral stability analysis of cantilever tapered steel beams. Different examples of nonlocal models and numerical methods can be found in the literature for a large variety of coupled problems and engineering applications. In Refs. [48,49], the authors proposed a Timoshenko beam nonlocal model to assess the free vibrational response of magneto-electro-elastic nanobeams [48], also made of FGMs [49]. The von Kármán geometric nonlinearity was included within a first-order shear deformable beam model by Liu et al. [50] in a nonlocal elasticity context, to evaluate the buckling and post-buckling responses of nanobeams made of piezoelectric materials in thermo-electro-mechanical conditions. A third-order shear deformable beam theory was adopted by Nami et al. [51] for a thermal stability analysis of FG nanoplates. Among tapered member applications, a novel beam finite element was introduced by Mohri et al. [52], together with a large torsion assumption, to estimate the stability resistance of tapered thin-walled beams. A semi-analytical procedure based on the Ritz technique was employed by Kuś [53] for analyzing the lateral stability of linearly tapered-web and/or flange doubly-symmetric I-beams. A finite element-based solution was proposed by Pandeya and Singhb [54] to survey the free vibrational behavior of a fixed–free nanobeam with a varying cross-section. According to the Eringen nonlocal theory and Euler–Bernoulli beam model, the nonlinear vibration of AFG nanobeams with a tapered section was exploited by Shafiei et al. [55], and a semi-analytical finite strip procedure was implemented by Zhang et al. [56,57] for the study of the stability capacity of bars with an open and closed cross-section under an axial loading condition [56], accounting for the effect of lateral elastic braces on the overall stability response in Ref. [57]. Further studies on the nonlocal vibration, buckling, and post-buckling of size-dependent beams, rods and plates at different scales can be found in [58,59,60,61,62,63,64,65,66,67], both in an analytical and a numerical sense. More specifically, as far as thin-walled structures are concerned, novel efficient models and computational methods have been developed in the literature to treat even more complicated applications. Among the most recent works, a novel optimization methodology was proposed by Maalawi [68] to enhance the vibrational response of thin-walled box beams with varying material properties. An innovative finite element formulation was also suggested by Lezgy-Nazargah [69] based on the theory of a generalized layered global–local beam (GLGB), to carry out an elasto-plastic analysis of thin-walled beams with reduced computational effort. Nguyen et al. [70,71] derived an efficient finite element formulation to investigate the flexural–torsional stability and buckling response of FGM beams with a singly symmetric open section, in the framework of Vlasov’s theory. Li et al. [72] applied the method of generalized differential quadrature to rigorously solve the bending, buckling and vibrational problems of AFG beams, accounting for nonlocal strain gradient theoretical assumptions. Moreover, Khaniki et al. [73,74,75,76,77,78,79,80] published several important contributions elated to the static, vibrational and buckling analysis of small-size beams with a constant or variable cross-section, made of homogenous and/or FGMs. A finite element approach was recently developed by Koutoati [81] to assess the static and free vibrations of multilayer composites and FG beams by means of different shear deformation beam theories. Following the first-order shear deformation theory, Glabisz et al. [82] formulated an innovate algorithm to analyze the stability and vibrational problem of nanobeams incorporating different end supports. Within a modified shear deformation theory context, in which it is not essential to use the shear correction factor, the stability and free vibration behavior of FG nanobeams were explored by Ebrahim et al. [83] using the Chebyshev–Ritz method. A double analytical and finite element solution has recently been proposed by Jrad et al. [84] to assess the triply coupled free vibrational responses of thin-walled beams under different boundary conditions. More recently, a third-order shear deformation theory was employed by Arefi and Civalek [85] to check for the static deformation of cylindrical nanoshells made from FG piezoelectric materials supported by a Pasternak elastic foundation.

Among the studies on tapered structures, Osmani and Meftah [86] studied the shear deformation effect on the buckling response of tapered I-shape beams under different loading conditions. An innovative methodology based on the classical energy approach was expounded by Chen et al. [87] for predicting the lateral buckling resistance of I-beams with simple supports. Achref et al. [88] analytically assessed the higher-order instability loads of beams with thin-walled open cross-sections under different loading conditions by resorting to a classical finite element approach for comparative purposes. Different numerical approaches were applied in Refs. [89,90,91,92,93] for the linear stability and free vibrational study of homogenous and AFG tapered thin-walled beams with an open cross-section, subjected to different boundary conditions and arbitrary loading cases. 

Based on the available literature, however, it seems that the flexural–torsional stability of AFG nanobeam-columns with tapered I-section has never been assessed. The current research is moving in this direction, and is aimed at probing the size-dependent buckling properties of AFG tapered nanobeams with a doubly-symmetric thin-walled cross-section, according to Vlasov assumptions. All the mechanical properties in the present work are graded in the longitudinal direction using the power function except, for the Poisson’s ratio, wherein the small size effect is taken into account via the Eringen nonlocal elasticity theory. The nonlocal governing equations of the problem, together with the associated boundary conditions, are obtained by implementing the Vlosov model and the energy method, in order to account for the eccentricity effect of of a compressive axial loading from the centroid within the formulation. The DQM is here employed to solve the resulting stability equations in a strong form and to determine the flexural–torsional buckling load. Different numerical examples analyze the effects of several parameters, namely, the constituent volume fractions, tapering ratio, nonlocal parameter and mode number, on the flexural–torsional stability of AFG tapered nanobeams with an I-section subjected to simply supported boundary conditions. The work is organized as follows. After a preliminary description of the theoretical formulation (Section 2), we provide (in Section 3) the basic notions of the DQM, here applied as an efficient tool to solve the problem with reduced computational effort. In Section 4 we present the results from a large parametric investigation aimed at checking the sensitivity of the mechanical response to different input parameters, which is useful for design purposes. The main results and concluding remarks are discussed in Section 5.

## 2. Problem Definition

The following stability model represents an extension of the formulation proposed in Ref. [94] for non-prismatic thin-walled nanobeam-columns with an arbitrary distribution of the material properties in the axial direction, whose numerical outcomes could be useful for the development and design of thin-walled structures, such as scanning tunneling microscopes with nonuniform nanobeams at tunneling tips. Due to the rapid development of nanoscience, the stability of FG nanobeams with variable thin-walled cross sections represents one of their key design benefits, as here explored theoretically via nonconventional Eringen nonlocal elasticity, and numerically via the DQM.

### 2.1. Kinematics 

Consider a straight tapered doubly symmetric I-beam made of non-homogeneous material, with variable properties along its longitudinal direction, as represented in Figure 1.

The orthogonal right-hand Cartesian coordinate system
$\left(x,y,z\right)$ is adopted, wherein *x* denotes the longitudinal axis, and y and z are the first and second principal bending axes parallel to the flanges and web, respectively. The origin *O* of these axes is located at the centroid of the cross-section. In the current work, it is assumed that the height of the web and/or width of both flanges can vary linearly along the longitudinal direction (x-axis), while the thickness remains constant. In the case of doubly-symmetric thin-walled sections, the shear center coincides with the centroid. In this study, we consider only slender beams, such that shear deformations can be ignored in our formulation, together with the local and distortional deformations. Based on these assumptions and following the Vlasov model for non-uniform torsion [95], the displacement field for an arbitrary point on the beam can be expressed as
(1a)$$U(x,y,z)=u(x)-y\frac{\partial v(x)}{\partial x}-z\frac{\partial w(x)}{\partial x}-\omega (y,z)\frac{\partial \theta (x)}{\partial x}$$
(1b)V(x,y,z)=v(x)−zθ(x)
(1c)W(x,y,z)=w(x)+yθ(x)

In these equations, U is the axial displacement, *V* and *W* represent the lateral and vertical displacements (along the *y*- and z-directions, respectively); *u*,*v*,*w* are the kinematic quantities defined at the reference surface; ω(y,z) stands for the warping function for the variable cross-section, which can be defined based on St. Venant torsional theory, and θ is the twisting angle. The Green strain tensor components in the large displacement include both the linear and the nonlinear strain parts, as follows
(2)$$\varepsilon_{ij}=\frac{1}{2}(\frac{\partial U_{i}}{\partial x_{j}}+\frac{\partial U_{j}}{\partial x_{i}})+\frac{1}{2}\left(\frac{\partial U_{k}}{\partial x_{i}}\frac{\partial U_{k}}{\partial x_{j}}\right)=\varepsilon_{ij}^{l}+\varepsilon_{ij}^{*}\;\;\;\;\;i,j,k=x,y,z$$
where $\varepsilon_{ij}^{l}$ denotes the linear part, and $\varepsilon_{ij}^{*}$ refers to the quadratic nonlinear part. For thin-walled beams, the strain tensor components reduce to the following: (3a)$$\varepsilon_{xx}\approx U^{\prime }+\frac{1}{2}\left(V\prime^{2}+W\prime^{2}\right)=\varepsilon_{xx}^{l}+\varepsilon_{xx}^{*}$$
(3b)$$\varepsilon_{xy}=\frac{1}{2}\left(\frac{\partial U}{\partial y}+\frac{\partial V}{\partial x}\right)+\frac{1}{2}\left(\frac{\partial V}{\partial x}\frac{\partial V}{\partial y}+\frac{\partial W}{\partial x}\frac{\partial W}{\partial y}\right)=\varepsilon_{xy}^{l}+\varepsilon_{xy}^{*}$$
(3c)$$\varepsilon_{xz}=\frac{1}{2}\left(\frac{\partial U}{\partial z}+\frac{\partial W}{\partial x}\right)+\frac{1}{2}\left(\frac{\partial V}{\partial x}\frac{\partial V}{\partial z}+\frac{\partial W}{\partial x}\frac{\partial W}{\partial z}\right)=\varepsilon_{xz}^{l}+\varepsilon_{xz}^{*}$$

By using Equations (1)–(3) and considering a tapering geometry, the non-zero linear and nonlinear parts of the strain displacement field are defined as
(4a)$$\varepsilon_{xx}^{l}=u_{}^{\prime }-yv^{\prime \prime }-zw^{\prime \prime }-\omega \theta^{\prime \prime }$$
(4b)$$\gamma_{{}_{xz}}^{l}=2\varepsilon_{{}_{xz}}^{l}=\left(y-\frac{\partial \omega }{\partial z}\right)\theta^{\prime }$$
(4c)$$\gamma_{{}_{xy}}^{l}=2\varepsilon_{{}_{xy}}^{l}=-\left(z+\frac{\partial \omega }{\partial y}\right)\theta^{\prime }$$
(4d)$$\varepsilon_{xx}^{*}=\frac{1}{2}\left[v\prime^{2}+w\prime^{2}+r^{2}\theta \prime^{2}\right]+yw\prime \theta \prime -zv\prime \theta \prime$$
(4e)$$\gamma_{xz}^{*}=-\left({v^{\prime }}_{}+\theta^{\prime }z\right)\theta$$
(4f)$$\gamma_{xy}^{*}=\left({w^{\prime }}_{}+\theta^{\prime }y\right)\theta$$
where $r^{2}=y^{2}+z^{2}$. In this study, we consider a compressive axial load *P* acting at the end of the beam along the *z*-direction, together with an external bending moment acting around the major principal axis, $M_{y}^{*}$, while assuming a null bending moment $M_{z}^{*}$ with respect to the z-axis. The most common cases of normal and shear stress associated with the external bending moment $M_{y}^{*}$ and shear force $V_{z}$ are considered as
(5a)$$\sigma_{xx}^{0}=\frac{P}{A}-\frac{M_{y}^{*}}{I_{y}}z$$
(5b)$$\tau_{xz}^{0}=\frac{V_{z}}{A}=-\frac{M_{y}^{*\prime }}{A}$$
where $\tau_{xz}^{0}$ is the mean value of the shear stress, $\sigma_{xx}^{0}$ stands for the initial normal stress in the cross-section, and A and $I_{y}$ are the cross-sectional area and second moment of inertia around the y-axis, defined as follows:
(6a)$$A=\int_{A}dA$$
(6b)$$I_{y}=\int_{A}z^{2}dA$$

### 2.2. Constitutive Relations

According to the Eringen nonlocal elasticity model [4], the stress at a point inside a body depends not only on the strain state at that point, but also on the strain states at all other points throughout the body. For homogenous and isotropic elastic solids, the nonlocal stress tensor σ at point x can be defined as
(7)$$\sigma_{ij}(x)=\int_{V}\alpha (\left|x^{\prime }-x\right|,\tau )C_{ijkl}\varepsilon_{kl}(x^{\prime })dV(x^{\prime })$$
where $\varepsilon_{kl}$ and $C_{ijkl}$ denote the linear strain components and the elastic stiffness coefficients, respectively. In addition, α(|x′−x|,τ) is the nonlocal kernel function, |x′−x| is the Euclidean distance, $\tau =e_{0}a/l$ stands for the material parameter, where a is an internal characteristic length (e.g., lattice parameter, *C*–*C* bond length or granular distance) and *l* is an external characteristic length in the nanostructures (e.g., crack length, wavelength), and *e*_0_ is a material constant, which is determined experimentally or in an approximate form by matching the dispersion curves of plane waves with those based on atomic lattice dynamics.

It is possible to express the integral constitutive equation presented in Equation (7) in the following differential constitutive equation:(8)$$\sigma_{ij}-\mu \nabla^{2}\sigma_{ij}=C_{ijkl}\varepsilon_{kl}$$
where $\nabla^{2}$ is the Laplacian operator and $\mu ={\left(e_{0}a\right)}^{2}$ stands for the nonlocal parameter. For a nonlocal AFG I-beam, the nonlocal constitutive relations can be written as
(9a)$$\sigma_{xx}-\mu \frac{\partial^{2}\sigma_{xx}}{\partial x^{2}}=E\varepsilon_{xx}^{l}$$
(9b)$$\tau_{xy}-\mu \frac{\partial^{2}\tau_{xy}}{\partial x^{2}}=G\gamma_{xy}^{l}$$
(9c)$$\tau_{xz}-\mu \frac{\partial^{2}\tau_{xz}}{\partial x^{2}}=G\gamma_{xz}^{l}$$
where *E* and *G* are the elastic and shear moduli, respectively, and $\sigma_{xx}$, $\tau_{xy}$, and $\tau_{xz}$ denote the Piola–Kirchhoff stress tensor components.

### 2.3. Equilibrium Equations

The principle of minimum total potential energy is applied to obtain the equilibrium equations together with the boundary conditions. For thin-walled beams, the total potential energy Π is expressed in its variational form by means of the elastic strain energy $U_{l}$ and the strain energy due to initial stress $U_{0}$,
(10)$$\delta \Pi =\delta \left(U_{l}+U_{0}\right)=0$$

Note that in a linear stability context, in the absence of an external force, the external work associated with the applied loads $W_{e}$ is equal to zero. At the same time, the variational form of the strain energy $\delta U_{l}$ is defined as
(11)$$\delta U_{l}=\int_{0}^{L}\int_{A}\left(\sigma_{xx}\delta \varepsilon_{xx}^{l}+\tau_{xy}\delta \gamma_{xy}^{l}+\tau_{xz}\delta \gamma_{xy}^{l}\right)dAdx$$
where L and *A* stand for the element length and cross-sectional area, respectively, and $\delta \varepsilon_{xx}^{l}$, $\delta \gamma_{xz}^{l}$ and $\delta \gamma_{xy}^{l}$ are the linear parts of the strain tensor in a variational form. By substituting Equation (4a–c) into Equation (11), the virtual elastic strain energy becomes
(12)$$\delta U_{l}=\int_{0}^{L}\int_{A}\sigma_{xx}\left(\delta u_{0}^{\prime }-y\delta v^{\prime \prime }-z\delta w^{\prime \prime }-\omega \delta \theta^{\prime \prime }\right)dAdx+\int_{0}^{L}\int_{A}\tau_{xy}\left(-(z+\frac{\partial \omega }{\partial y})\delta \theta^{\prime }\right)dAdx+\int_{0}^{L}\int_{A}\tau_{xz}\left((y-\frac{\partial \omega }{\partial z})\delta \theta^{\prime }\right)dAdx$$

By integration over the cross-sectional area, we get
(13)$$\delta U_{l}=\int_{L}\left(N\delta u_{0}^{\prime }+M_{z}\delta v^{\prime \prime }-M_{y}\delta w^{\prime \prime }+B_{\omega }\delta \theta^{\prime \prime }\right)dx+\int_{0}^{L}\left(M_{sv}\delta \theta^{\prime }\right)dx$$
where N is the axial force, $M_{y}$ and $M_{z}$ denote the two bending moments, $B_{\omega }$ is the bi-moment, and $M_{sv}$ is the St. Venant torsional moment. These stress resultants in Equation (13) are defined as
(14a)$$N=\int_{A}\sigma_{xx}dA$$
(14b)$$M_{y}=\int_{A}\sigma_{xx}zdA$$
(14c)$$M_{z}=-\int_{A}\sigma_{xx}ydA$$
(14d)$$B_{\omega }=-\int_{A}\sigma_{xx}\omega dA$$
(14e)$$M_{sv}=\int_{A}\left(\tau_{xz}(y-\frac{\partial \omega }{\partial z})-\tau_{xy}(z+\frac{\partial \omega }{\partial y})\right)dA$$

Moreover, the variation in the strain energy due to the initial stresses can be stated as
(15)$$\delta U_{0}=\int_{0}^{L}\int_{A}\left(\sigma_{xx}^{0}\delta \varepsilon_{xx}^{*}+\tau_{xy}^{0}\delta \gamma_{xy}^{*}+\tau_{xz}^{0}\delta \gamma_{xz}^{*}\right)dAdx$$

By introducing the first variation in the nonlinear strain-displacement relations, defined by Equation (4d–f), and the initial stresses (5a,b) in Equation (15), we get the following relation:(16)$$\begin{array}{l} \delta U_{0}=\int_{0}^{L}\int_{A}(\frac{P}{A}-\frac{M_{y}^{*}}{I_{y}}z)\left(v^{\prime }\delta v^{\prime }+w^{\prime }\delta w^{\prime }+r^{2}\theta^{\prime }\delta \theta^{\prime }+y\theta \prime \delta w\prime +yw\prime \delta \theta \prime -z\theta \prime \delta v\prime -zv\prime \delta \theta \prime \right)dAdx\\ +\int_{0}^{L}\int_{A}(-\frac{M_{y}^{*\prime }}{A})\left(-\theta \delta v^{\prime }-v^{\prime }\delta \theta -z\theta \delta \theta^{\prime }-z\theta^{\prime }\delta \theta \right)dAdx \end{array}$$

At this stage, by integrating Equation (16) over the cross-section, the variation in the strain energy due to the initial stresses takes the following final form:(17)$$\delta U_{0}=\int_{0}^{L}\left(P(v^{\prime }\delta v^{\prime }+w^{\prime }\delta w^{\prime }+\frac{I_{y}+I_{z}}{A}\theta^{\prime }\delta \theta^{\prime })\right)dx+\int_{0}^{L}\left(M_{y}^{*}(\theta \prime \delta v\prime +v\prime \delta \theta \prime )\right)dx+\int_{0}^{L}\left(M_{y}^{*\prime }(\theta \delta v^{\prime }+v^{\prime }\delta \theta )\right)dx$$
or equivalently
(18)$$\delta U_{0}=\int_{0}^{L}\left(Pv^{\prime }\delta v^{\prime }+Pw^{\prime }\delta w^{\prime }+r_{O}^{2}\theta^{\prime }\delta \theta^{\prime }\right)dx+\int_{0}^{L}\left(-M_{y}^{*}v^{\prime \prime }\delta \theta -M_{y}^{*}\theta \delta v^{\prime \prime }\right)dx$$

In Equation (18), $I_{z}$ is the second moment of inertia around the z-axis and $r_{0}$ is the polar radius gyration around the centroid, given by
$$I_{z}=\int_{A}y^{2}dA,\;r_{o}=\sqrt{\frac{I_{y}+I_{z}}{A}}$$

By introducing Equations (13) and (18) into Equation (10) and setting the coefficients of $\delta u_{0},\delta v,\delta w,\delta \theta ,$ as to zero, we obtain the equilibrium equations
(19a)$$N^{\prime }=0$$
(19b)$$-M_{y}^{\prime \prime }-(Pw^{\prime }{)}^{\prime }=0$$
(19c)$$M_{z}^{\prime \prime }-(M_{y}^{*}\theta {)}^{\prime \prime }-(Pv^{\prime }{)}^{\prime }=0$$
(19d)$$M_{\omega }^{\prime \prime }-M_{y}^{*}v^{\prime \prime }-M_{sv}^{\prime }-(Pr_{O}^{2}\theta^{\prime }{)}^{\prime }=0$$
under the following boundary conditions
(20)$$N=0\;\;\;\;\;\mathrm{or}\;\;\;\;\;\delta u_{0}=0\;-M_{y}=0\;\;\;\;\;\mathrm{or}\;\;\;\;\;\delta w^{\prime }=0\;M_{y}^{\prime }+Pw^{\prime }=0\;\;\;\;\;\mathrm{or}\;\;\;\;\;\delta w=0\;M_{z}-M_{y}^{*}\theta =0\;\;\;\;\;\mathrm{or}\;\;\;\;\;\delta v\prime =0\;-M_{z}^{\prime }+(M_{y}^{*}\theta {)}^{\prime }+Pv^{\prime }=0\;\;\;\;\;\mathrm{or}\;\;\;\;\;\delta v=0\;-B_{\omega }=0\;\;\;\;\;\mathrm{or}\;\;\;\;\;\delta \theta^{\prime }=0\;B_{\omega }^{\prime }+M_{sv}+Pr_{O}^{2}\theta^{\prime }=0\;\;\;\;\;\mathrm{or}\;\;\;\;\;\delta \theta =0$$

By substituting Equation (4a–c) into Equation (9) and the subsequent results into Equation (14), the stress resultants are obtained as
(21a)$$N-\mu \frac{\partial^{2}N}{\partial x^{2}}=ΕAu_{0}{}^{\prime }$$
(21b)$$M_{y}-\mu \frac{\partial^{2}M_{y}}{\partial x^{2}}=-ΕI_{y}w^{\prime \prime }$$
(21c)$$M_{z}-\mu \frac{\partial^{2}M_{z}}{\partial x^{2}}=ΕI_{z}v^{\prime \prime }$$
(21d)$$B_{\omega }-\mu \frac{\partial^{2}B_{\omega }}{\partial x^{2}}=ΕI_{\omega }\theta^{\prime \prime }$$
(21e)$$M_{sv}-\mu \frac{\partial^{2}M_{sv}}{\partial x^{2}}=GJ\theta^{\prime }$$

In the previous expressions, J and $I_{\omega }$ are the St. Venant torsion and warping constants, defined as
(22a)$$\;I_{\omega }=\int_{A}\omega^{2}dA,$$
(22b)$$J=\int_{A}\left({\left(y-\frac{\partial \omega }{\partial z}\right)}^{2}+{\left(z+\frac{\partial \omega }{\partial y}\right)}^{2}\right)dA$$

This study is established in the context of small displacements and deformations. According to the linear stability, the nonlinear terms are also disregarded in the equilibrium equations. Based on these assumptions, the system of equilibrium equations for tapered I-beams under a nonlocal theory are finally derived by placing Equation (21) into Equation (19)
(23a)$${\left(EA{u^{\prime }}_{0}\right)}^{\prime }=0$$
(23b)$$(EI_{y}w^{\prime \prime }{)}^{\prime \prime }+\mu (Pw^{\prime }{)}^{\prime \prime \prime }-(Pw^{\prime }{)}^{\prime }=0$$
(23c)$$(EI_{z}v^{\prime \prime }{)}^{\prime \prime }+\mu (M_{y}^{*}\theta {{)}^{\prime \prime \prime }}^{\prime }-(M_{y}^{*}\theta {)}^{\prime \prime }+\mu (Pv^{\prime }{)}^{\prime \prime \prime }-(Pv^{\prime }{)}^{\prime }=0$$
(23d)$$(EI_{\omega }\theta^{\prime \prime }{)}^{\prime \prime }-(GJ\theta^{\prime }{)}^{\prime }+\mu (M_{y}^{*}v^{\prime \prime }{)}^{\prime \prime }-M_{y}^{*}v^{\prime \prime }+\mu (Pr_{O}^{2}\theta^{\prime }{)}^{\prime \prime \prime }-(Pr_{O}^{2}\theta^{\prime }{)}^{\prime }=0$$

The related boundary conditions at the ends of the thin-walled nanobeam can be expressed as
(24)$${\left(EA{u^{\prime }}_{0}\right)}^{\prime }=0\;\mathrm{or}\;\delta u_{0}=0$$
$$EI_{y}w^{\prime \prime }=0\;\mathrm{or}\;\delta w^{\prime }=0$$
$$-(EI_{y}w^{\prime \prime }{)}^{\prime }-\mu (Pw^{\prime }{)}^{\prime \prime }+Pw^{\prime }=0\;\mathrm{or}\;\delta w=0$$
$$EI_{z}v^{\prime \prime }-M_{y}^{*}\theta +\mu (M_{y}^{*}\theta {)}^{\prime \prime }=0\;\mathrm{or}\;\delta v^{\prime }=0$$
$$-(ΕI_{z}v^{\prime \prime }{)}^{\prime }-\mu (M_{y}^{*}\theta {)}^{\prime \prime \prime }-\mu (Pv^{\prime }{)}^{\prime \prime }+(M_{y}^{*}\theta {)}^{\prime }+Pv^{\prime }=0\;\mathrm{or}\;\delta v=0$$
$$EI_{\omega }\theta^{\prime \prime }=0\;\mathrm{or}\;\delta \theta^{\prime }=0$$
$$-(EI_{\omega }\theta^{\prime \prime }{)}^{\prime }+GJ\theta^{\prime }-\mu (M_{y}^{*}v^{\prime \prime }{)}^{\prime }+M_{y}^{*}v^{\prime }-\mu (Pr_{O}^{2}\theta^{\prime }{)}^{\prime \prime }+(Pr_{O}^{2}\theta^{\prime })=0\;\mathrm{or}\;\delta \theta =0$$

In the following section, a numerical solution procedure based on the DQM is applied to solve the governing equations for the flexural–torsional buckling of AFG nanobeams with varying I-sections, as has been successfully carried out in the literature for a large variety of problems [96,97,98,99,100,101,102,103].

## 3. Numerical Solution Method 

Due to the varying cross-sectional mechanical properties, the resulting flexural–torsional stability Equation (23a–d) for I-tapered nanobeams represent a system of three-coupled fourth-order differential equations with variable coefficients. Under these conditions, it is not possible to accurately estimate a general and straightforward closed-form solution. For such complicated problems, the DQM-based approach, as proposed for the first time by Bellman and Casti [96], is here employed as an efficient and easy tool to solve the coupled differential equations of the problem in a strong form. The basic concept of the proposed method relies on the possibility of discretizing the derivatives of a function with respect to a variable in differential equations at some fixed collocation points by means of a weighted linear summation of the function’s values at its adjacent points. The governing equations, together with the associated boundary conditions, are thus transformed into a set of linear algebraic equations, which can be solved with the aid of a computational algorithm to derive an approximate solution for continuous differential equations. To this end, it is necessary to divide the computational region into a fixed number of grid points spanning the solution domain. The accuracy of this numerical approach depends on the number and types of selected sampling points, as also discussed in Refs. [97,98,99,100,101,102,103]. One of the best options for the sampling points in the stability and vibration analysis is the Chebyshev–Gauss–Lobatto points:(25)$$x_{i}=\frac{L}{2}\left[1-\cos\left(\frac{i-1}{N-1}\pi \right)\right],\;\mathrm{if}\;0\leq x\leq L\;i=1,2,\ldots ,N$$
where N is the total number of grid points in the longitudinal direction. According to DQM, the $m^{\mathrm{th}}$-order derivative of a function f(ξ) at a fixed grid point $\xi_{i}$ can be approximated as
(26)$${\frac{d^{m}f}{d\xi^{m}}|}_{\xi =\xi_{i}}=\sum_{j=1}^{N}A_{ij}^{\left(m\right)}f(\xi_{j})\;\;\;for\;\;\;i=1,2,\ldots ,N$$
where $f(\xi_{j})$ refers to the functional value at grid points $\xi_{j}\;\;(i=1,2,\ldots ,N)$, and $A_{ij}^{\left(m\right)}$ is the weighting coefficient for the $m^{\mathrm{th}}$-order derivative. The first-order derivative of the weighting coefficient $A_{ij}^{\left(1\right)}$ is computed by the following algebraic formulation based on the Lagrangian interpolation polynomials,
(27)$$A_{ij}^{\left(1\right)}=\left\{ \begin{array}{l} \frac{M(\xi_{i})}{\left(\xi_{i}-\xi_{j})M(\xi_{j}\right)}\;\;for\;\;i\neq j\\ -\sum_{k=1,k\neq i}^{N}A_{ik}^{\left(1\right)}\;\;for\;\;i=j \end{array}\right.\;\;i,j=1,2,\ldots ,N$$
where
(28)$$M(\xi_{i})=\prod_{j=1,j\neq i}^{N}\left(\xi_{i}-\xi_{j}\right)\;\;for\;\;i=1,2,\ldots ,N$$

The higher-order DQM weighting coefficients can be acquired from the first-order ones, as follows:(29)$$A_{ij}^{\left(m\right)}=A_{ij}^{\left(1\right)}A_{ij}^{\left(m-1\right)}\;\;\;\;\;\;\;\;2\leq m\leq N-1$$

In order to solve the stability equation by means of the differential quadrature approach, a dimensionless variable (ξ=x/L) is introduced. By the expansion of Equation (23), the governing equations of the problem take the following final discrete form:(30a)$$\begin{array}{l} Ε(\xi_{j})I_{y}(\xi_{j})(\sum_{j=1}^{N}A_{ij}^{\left(4\right)}w_{j})+2(Ε(\xi_{j})I_{y}^{\prime }(\xi_{j})+{Ε}^{\prime }(\xi_{j})I_{y}(\xi_{j}))(\sum_{j=1}^{N}A_{ij}^{\left(3\right)}w_{j})\\ +({Ε}^{\prime \prime }(\xi_{j})I_{y}(\xi_{j})+2{Ε}^{\prime }(\xi_{j})I_{y}^{\prime }(\xi_{j})+E(\xi_{j})I_{y}^{\prime \prime }(\xi_{j}))(\sum_{j=1}^{N}A_{ij}^{\left(2\right)}w_{j})+\mu P(\sum_{j=1}^{N}A_{ij}^{\left(4\right)}w_{j})-L^{2}P(\sum_{j=1}^{N}A_{ij}^{\left(2\right)}w_{j})=0 \end{array}$$
(30b)$$\begin{array}{l} Ε(\xi_{j})I_{z}(\xi_{j})(\sum_{j=1}^{N}A_{ij}^{\left(4\right)}v_{j})+2(Ε(\xi_{j})I_{z}^{\prime }(\xi_{j})+{Ε}^{\prime }(\xi_{j})I_{z}(\xi_{j}))(\sum_{j=1}^{N}A_{ij}^{\left(3\right)}v_{j})\\ +({Ε}^{\prime \prime }(\xi_{j})I_{z}(\xi_{j})+2{Ε}^{\prime }(\xi_{j})I_{z}^{\prime }(\xi_{j})+E(\xi_{j})I_{z}^{\prime \prime }(\xi_{j}))(\sum_{j=1}^{N}A_{ij}^{\left(2\right)}v_{j})+\mu P(\sum_{j=1}^{N}A_{ij}^{\left(4\right)}v_{j})-L^{2}P(\sum_{j=1}^{N}A_{ij}^{\left(2\right)}v_{j})\\ +\mu M_{y}^{*}(\xi_{j})(\sum_{j=1}^{N}A_{ij}^{\left(4\right)}\theta_{j})+4\mu M_{y}^{*\prime }(\xi_{j})(\sum_{j=1}^{N}A_{ij}^{\left(3\right)}\theta_{j})+(6\mu M_{y}^{*\prime \prime }(\xi_{j})-L^{2}M_{y}^{*}(\xi_{j}))(\sum_{j=1}^{N}A_{ij}^{\left(2\right)}\theta_{j})\\ +(4\mu M_{y}^{*\prime \prime \prime }(\xi_{j})-2L^{2}M_{y}^{*\prime }(\xi_{j}))(\sum_{j=1}^{N}A_{ij}^{\left(1\right)}\theta_{j})+(\mu M_{y}^{*4}(\xi_{j})-L^{2}M_{y}^{*\prime \prime }(\xi_{j}))\theta_{j}=0 \end{array}$$
(30c)$$\begin{array}{l} Ε(\xi_{j})I_{\omega }(\xi_{j})(\sum_{j=1}^{N}A_{ij}^{\left(4\right)}\theta_{j})+2(Ε(\xi_{j})I_{\omega }^{\prime }(\xi_{j})+{Ε}^{\prime }(\xi_{j})I_{\omega }(\xi_{j}))(\sum_{j=1}^{N}A_{ij}^{\left(3\right)}\theta_{j})\\ +({Ε}^{\prime \prime }(\xi_{j})I_{\omega }(\xi_{j})+2{Ε}^{\prime }(\xi_{j})I_{\omega }^{\prime }(\xi_{j})+E(\xi_{j})I_{\omega }^{\prime \prime }(\xi_{j})-L^{2}G(\xi_{j})J(\xi_{j}))(\sum_{j=1}^{N}A_{ij}^{\left(2\right)}\theta_{j})\\ -L^{2}(G^{\prime }(\xi_{j})J(\xi_{j})+G(\xi_{j})J^{\prime }(\xi_{j}))(\sum_{j=1}^{N}A_{ij}^{\left(1\right)}\theta_{j})\\ +\mu PR_{c}(\xi_{j})(\sum_{j=1}^{N}A_{ij}^{\left(4\right)}\theta_{j})+3\mu PR_{o}^{\prime }(\xi_{j})(\sum_{j=1}^{N}A_{ij}^{\left(3\right)}\theta_{j})+P(3\mu R_{c}^{\prime }(\xi_{j})-L^{2}R_{o}(\xi_{j}))(\sum_{j=1}^{N}A_{ij}^{\left(2\right)}\theta_{j})\\ +P(\mu R_{o}^{\prime \prime }(\xi_{j})-L^{2}R_{o}^{\prime }(\xi_{j}))(\sum_{j=1}^{N}A_{ij}^{\left(1\right)}\theta_{j})+\mu M_{y}^{*}(\xi_{j})(\sum_{j=1}^{N}A_{ij}^{\left(4\right)}v_{j})+2\mu M_{y}^{*\prime }(\xi_{j})(\sum_{j=1}^{N}A_{ij}^{\left(3\right)}v_{j})\\ +(\mu M_{y}^{*\prime \prime }(\xi_{j})-L^{2}M_{y}^{*}(\xi_{j}))(\sum_{j=1}^{N}A_{ij}^{\left(2\right)}v_{j})=0 \end{array}$$
where $R_{o}=r_{o}^{2}$ in Equation (30c).

By rewriting the problem in matrix form, we get the following relation,
(31)$$\left({\left[\begin{array}{ccc} \left[K_{ww}\right] & \left[0\right] & \left[0\right]\\ \left[0\right] & \left[K_{vv}\right] & \left[0\right]\\ \left[0\right] & \left[0\right] & \left[K_{\theta \theta }\right] \end{array}\right]}_{3N\times 3N}+{\left[\begin{array}{ccc} \left[P_{ww}\right] & \left[0\right] & \left[0\right]\\ \left[0\right] & \left[P_{vv}\right] & \left[0\right]\\ \left[0\right] & \left[0\right] & \left[P_{\theta \theta }\right] \end{array}\right]}_{3N\times 3N}+{\left[\begin{array}{ccc} \left[0\right] & \left[0\right] & \left[0\right]\\ \left[0\right] & \left[0\right] & \left[M_{v\theta }\right]\\ \left[0\right] & \left[M_{\theta v}\right] & \left[0\right] \end{array}\right]}_{3N\times 3N}\right)\times {\left\{\begin{array}{c} \left\{w\right\}\\ \left\{v\right\}\\ \left\{\theta \right\} \end{array}\right\}}_{3N\times 1}={\left\{\begin{array}{c} \left\{0\right\}\\ \left\{0\right\}\\ \left\{0\right\} \end{array}\right\}}_{3N\times 1}$$
where
(32)$$\begin{array}{l} {}[K_{ww}]=\left[a^{1}\right]{\left[A\right]}^{\left(4\right)}+\left[b^{1}\right]{\left[A\right]}^{\left(3\right)}+\left[c^{1}\right]{\left[A\right]}^{\left(2\right)}\\ \left[P_{ww}]=P(\mu {\left[A\right]}^{\left(4\right)}-L^{2}{\left[A\right]}^{\left(2\right)}\right) \end{array}$$
$$\begin{array}{l} {}[K_{vv}]=\left[a^{2}\right]{\left[A\right]}^{\left(4\right)}+\left[b^{2}\right]{\left[A\right]}^{\left(3\right)}+\left[c^{2}\right]{\left[A\right]}^{\left(2\right)}\\ \left[P_{vv}]=P(\mu {\left[A\right]}^{\left(4\right)}-L^{2}{\left[A\right]}^{\left(2\right)}\right)\\ {}[M_{v\theta }]=\left[i^{2}\right]{\left[A\right]}^{\left(4\right)}+\left[j^{2}\right]{\left[A\right]}^{\left(3\right)}+\left[k^{2}\right]{\left[A\right]}^{\left(2\right)}+\left[l^{2}\right]{\left[A\right]}^{\left(1\right)}+\left[m^{2}\right] \end{array}$$
$$\begin{array}{l} {}[K_{\theta \theta }]=\left[a^{3}\right]{\left[A\right]}^{\left(4\right)}+\left[b^{3}\right]{\left[A\right]}^{\left(3\right)}+\left[c^{3}\right]{\left[A\right]}^{\left(2\right)}-\left[d^{3}\right]{\left[A\right]}^{\left(1\right)}\\ \left[P_{\theta \theta }]=P(\left[e^{3}\right]{\left[A\right]}^{\left(4\right)}+\left[f^{3}\right]{\left[A\right]}^{\left(3\right)}+\left[g^{3}\right]{\left[A\right]}^{\left(2\right)}+\left[h^{3}\right]{\left[A\right]}^{\left(1\right)}\right)\\ {}[M_{\theta v}]=\left[i^{3}\right]{\left[A\right]}^{\left(4\right)}+\left[j^{3}\right]{\left[A\right]}^{\left(3\right)}+\left[k^{3}\right]{\left[A\right]}^{\left(2\right)} \end{array}$$
in which
(33)$$a_{jk}^{1}={(ΕI_{y}|}_{\xi =\xi_{j}})\delta_{jk};\;b_{jk}^{1}={\left(2(ΕI_{y}^{\prime }+{Ε}^{\prime }I_{y}\right)|}_{\xi =\xi_{j}})\delta_{jk};\;c_{jk}^{1}={\left(({Ε}^{\prime \prime }I_{y}+2{Ε}^{\prime }I_{y}^{\prime }+EI_{y}^{\prime \prime }\right)|}_{\xi =\xi_{j}})\delta_{jk}$$
$$\begin{array}{l} a_{jk}^{2}={(ΕI_{z}|}_{\xi =\xi_{j}})\delta_{jk};\;b_{jk}^{2}={\left(2(ΕI_{z}^{\prime }+{Ε}^{\prime }I_{z}\right)|}_{\xi =\xi_{j}})\delta_{jk};\;c_{jk}^{2}={\left(({Ε}^{\prime \prime }I_{z}+2{Ε}^{\prime }I_{z}^{\prime }+EI_{z}^{\prime \prime }\right)|}_{\xi =\xi_{j}})\delta_{jk}\\ i_{jk}^{2}={(\mu M_{y}^{*}|}_{\xi =\xi_{j}})\delta_{jk};j_{jk}^{2}={(4\mu M_{y}^{*\prime }|}_{\xi =\xi_{j}})\delta_{jk};k_{jk}^{2}={\left((6\mu M_{y}^{*\prime \prime }-L^{2}M_{y}^{*}\right)|}_{\xi =\xi_{j}})\delta_{jk}\\ l_{jk}^{2}={\left((4\mu M_{y}^{*\prime \prime }-2L^{2}M_{y}^{*\prime }\right)|}_{\xi =\xi_{j}})\delta_{jk};m_{jk}^{2}={\left((\mu M_{y}^{*}{}^{4}-L^{2}M_{y}^{*\prime \prime }\right)|}_{\xi =\xi_{j}})\delta_{jk} \end{array}$$
$$\begin{array}{l} a_{jk}^{3}={(ΕI_{\omega }|}_{\xi =\xi_{j}})\delta_{jk};\;b_{jk}^{3}={\left(2(ΕI_{\omega }^{\prime }+{Ε}^{\prime }I_{\omega }\right)|}_{\xi =\xi_{j}})\delta_{jk};\;c_{jk}^{3}={\left(({Ε}^{\prime \prime }I_{\omega }+2{Ε}^{\prime }I_{\omega }^{\prime }+EI_{\omega }^{\prime \prime }-L^{2}GJ\right)|}_{\xi =\xi_{j}})\delta_{jk};\\ d_{jk}^{3}={\left(L^{2}(G^{\prime }J+GJ^{\prime }\right)|}_{\xi =\xi_{j}})\delta_{jk};\;e_{jk}^{3}={P(\mu R_{o}|}_{\xi =\xi_{j}})\delta_{jk};\;f_{jk}^{3}={P(3\mu R_{o}^{\prime }|}_{\xi =\xi_{j}})\delta_{jk};\\ g_{jk}^{3}={P((3\mu R_{o}^{\prime }-L^{2}R_{o})|}_{\xi =\xi_{j}})\delta_{jk};\;h_{jk}^{3}={P((\mu R_{o}^{\prime \prime }-L^{2}R_{o}^{\prime \prime \prime })|}_{\xi =\xi_{j}})\delta_{jk}\\ i_{jk}^{3}={(\mu M_{y}^{*}|}_{\xi =\xi_{j}})\delta_{jk};\;j_{jk}^{3}={(2\mu M_{y}^{*\prime }|}_{\xi =\xi_{j}})\delta_{jk};\;k_{jk}^{3}={\left((\mu M_{y}^{*\prime \prime }-L^{2}M_{y}^{*}\right)|}_{\xi =\xi_{j}})\delta_{jk} \end{array}$$
and $\delta_{jk}$ is the Kronecker delta, defined as
(34)$$\delta_{jk}=\{\begin{array}{c} 0\;if\;j\neq k;\\ 1\;if\;j=k. \end{array}$$

In Equation (31), the displacement vectors and the torsion angle vector are defined as
(35)$$\begin{array}{l} {\left\{w\right\}}_{N\times 1}={\left\{\begin{array}{cccc} w_{1} & w_{2} & \ldots & w_{N} \end{array}\right\}}^{T};\;{\left\{v\right\}}_{N\times 1}={\left\{\begin{array}{cccc} v_{1} & v_{2} & \ldots & v_{N} \end{array}\right\}}^{T};\\ {\left\{\theta \right\}}_{N\times 1}={\left\{\begin{array}{cccc} \theta_{1} & \theta_{2} & \ldots & \theta_{N} \end{array}\right\}}^{T} \end{array}$$

The simple form of the final equation, Equation (31), can be stated as
(36)$${\left(\left[K\right]-\lambda (\left[P\right]+\left[M\right])\right)}_{3N\times 3N}{\left\{d\right\}}_{3N\times 1}={\left\{0\right\}}_{3N\times 1}$$
or
(37)$$\left(\left[K\right]-\lambda \left[K_{G}\right]\right)\left\{d\right\}=\{0\}$$
in which
(38a)$$\left[K_{G}]=[P]+[M\right]$$
(38b)$$\{d\}=\left\{\begin{array}{l} \left\{w\right\}\\ \left\{v\right\}\\ \left\{\theta \right\} \end{array}\right\}$$
[K] and $\left[K_{G}\right]$ are 3N×3N matrices, λ is the eigenvalues and {d} is the related eigenvectors. After the implementation of the boundary conditions, we compute the flexural–torsional buckling load from Equation (37), together with the associated vertical and lateral deflections and the twist angles of the AFG nanobeams.

## 4. Numerical Examples

In this section, we perform a parametric investigation to assess the sensitivity of the linear stability of AFG thin-walled nanobeam-columns (with a variable I-section and simply supported boundary conditions) to different material properties, as well as to different web and flange tapering parameters, mode numbers, nonlocal parameters, and axial load eccentricities. In what follows, we use the subscripts ${\left(•\right)}_{0}$ and ${\left(•\right)}_{1}$ to define the mechanical and geometrical properties of beams in their left $\left(x=0,\;\xi =0\right)$ and right $\left(x=L,\;\xi =1\right)$ supports, respectively. The dimensionless buckling load parameter is determined as
(39)$$P_{nor}=\frac{P_{cr}L^{2}}{E_{0}I_{Z0}}$$
which accounts for simply supported, tapered beams with I-sections subjected to a compressive axial force. In this regard, it is presumed that the widths of both flanges, $b_{0}$, and the web height, $d_{0}$, of the I-section on the left side increase linearly up to $b_{1}=(1+\beta )b_{0}$ and $d_{1}=(1+\alpha )d_{0}$ on the right side (Figure 2). Thus, the flanges and web tapering ratios are defined as $\beta =b_{1}/b_{0}-1$ and $\alpha =d_{1}/d_{0}-1$, respectively. Note that these two parameters $\left(\alpha ,\beta \right)$ are non-negative variables and can change simultaneously or separately. At the same time, by equating $\left(\alpha ,\beta \right)$ to zero, we revert to I-beams with a uniform cross-section. The geometrical schemes and dimensionless parameters are depicted in Figure 2. To perform the flexural–torsional buckling analysis, it is supposed that the compressive axial load is applied at three different positions: the top flange (TF) of the left side (i.e., for x=0), the centroid, and the TF of the right side (i.e., for x=L). 

For the same benchmark, the beams feature axially varying materials, ranging between pure ceramic on the side end and pure metal on the right side, according to a simple power-law function. More specifically, the ceramic phase is made of alumina $\left(Al_{2}O_{3}\right)$ with an equivalent Young’s modulus $E_{0}=380$ GPa, whereas the metal phase is aluminum $\left(\mathrm{Al}\right)$ with an equivalent Young’s modulus $E_{1}=70$ GPa, without considering the exact grain sizes and shapes of each material constituent. This means that the modulus of elasticity at an arbitrary coordinate is defined as
(40)$$E(\xi )=E_{0}+(E_{1}-E_{0})\xi^{m}$$
where the power-law index m assumes a positive value, and is zero only in a pure metal member.

We carry out a preliminary study aimed at defining the appropriate number of grid points within the domain to yield accurate results in terms of flexural–torsional buckling load. In the absence of further numerical nonlocal studies on the same thin-walled examples, the accuracy of our formulation is checked by comparing our results with predictions based on a classical finite element method, as performed via the commercial ANSYS code [104]. In detail, we evaluate the lowest values of the dimensionless buckling parameter $\left(P_{nor}\right)$ for the same structure made of pure alumina with three different loading positions and different tapering ratios, α=β=0÷1 by steps of 0.2 versus an increased number of grid points N. The main results are summarized in Table 1, where it seems that a total number of grid points N=20 is sufficient to obtain the lowest normalized buckling load for different axial load positions and non-uniformity parameters. Based on results in Table 1, we can observe the good agreement between our mathematical DQM-based formulation and predictions made via the ANSYS code [104] for each selected loading case. 

After the validation phase of the model, we continue with a systematic study of the flexural–torsional buckling of AFG nanobeams with different input parameters, such as eccentric axial load, web and flange non-uniformity parameters, gradient index, mode number and nonlocality parameter. In order to assess the linear stability strength of AFG nanobeams with varying I-sections, we compute the lowest normalized flexural–torsional buckling loads $\left(P_{nor}\right)$ of AFG tapered thin-walled nanobeams subjected to simply supported end conditions, as reported in Table 2, for different tapering ratios $\left(\alpha =\beta =0,\;\;0.3,\;\;0.6,\;\;0.9\right)$, material compositions (power-law exponent), nonlocal parameters $\left(\mu =0\;\;\mathrm{and}\;\;2\right)$, and three different loading positions. The compressive axial load can be applied on the TF of the left side $\left(x=0\right)$, at the centroid, and on the TF of the right side $\left(x=L\right)$. In Figure 3, Figure 4 and Figure 5, we represent the variation in $P_{nor}$ depending on Eringen’s nonlocal parameters (ranging from 0 to 3) for thin-walled beams with homogenous materials or an FG beam with different gradient indexes m=0.6,  1.3  and  2, while varying the tapering ratios from 0 to 0.9 and assuming three axial load positions, namely, on the TF for x=0, on the centroid, and the TF for x=L. In this case, we consider a non-uniform beam with equal web height and flange width tapering ratios, α=β. 

In Table 3, Table 4 and Table 5, we list the magnitude of the normalized flexural–torsional buckling parameter, $P_{nor}$, for various combinations of web height and flange width tapering ratio, β and α, and nonlocal parameters (μ = 0, 1 and 3) with different non-homogenous indices $\left(m=0.6,\;\;1.2\;\;\mathrm{and}\;\;1.8\right)$. The contribution of a possible axial load eccentricity at the cross-section centroid on the buckling resistance is also taken into account. The normalized buckling parameters are respectively illustrated in Table 3 and Table 4 for load positions on the TF at x=0 and on the shear center, as well as in Table 5 for a load position on the TF at x=L.

Under the first assumed load position, Figure 6 illustrates the variation in the normalized buckling load with respect to the web tapering ratio α and the flange tapering parameter β for different nonlocal parameters and material compositions, i.e., for a pure ceramic and AFG with m=1. The same analysis is repeated for an axial load located at x=L, whose results are plotted in Figure 7, where the reduction in the buckling load is observable with increasing values of μ and decreased tapering ratios of α and β. Moreover, in Figure 8, Figure 9 and Figure 10 we plot the lowest buckling load $P_{nor}$ versus the tapering ratio for different values of gradient indexes, m, and nonlocal parameters $\left(\mu =0,\;\;1,\;\;2\;\;\mathrm{and}\;\;3\right)$, while assuming α=β, and three different positions of compressive load, as in the previous cases. Note that a beam subjected to a compressive axial force on the centroid can buckle in a pure torsional buckling mode. In this context, Table 6 addresses the influence of web and flange tapering ratios, material composition (power-law exponent), and nonlocal parameters $\left(\mu =0,\;\;0.5\;\;\mathrm{and}\;\;1\right)$ on the normalized torsional buckling load of the tapered nanobeam. Based on the results for both local and nonlocal beams and all non-uniformity ratios, the stability strength improves as the non-homogeneity parameter increases. In other words, a higher flexural–torsional buckling capacity is obtained with an increased power index, m, due to the increased content of ceramic phase and increased gradient index, under a fixed ratio of α=β. As also visible in Figure 8, Figure 9 and Figure 10, the rate of increase in the critical load with α,β is gradual for increased values of m. Based on a comparative evaluation of the results in Figure 8, Figure 9 and Figure 10, with the same assumptions for α,β,m  and  μ, it seems that the load position has a significant effect on the stability strength of nanobeams with varying doubly symmetric I-sections, especially for higher values of the web tapering ratio. As also expected, the highest buckling capacity is obtained when the axial load is located exactly on the centroid, whereas the worst response corresponds to a compressive load applied on the TF owing to the compression of an initial bending moment resulting from a load eccentricity.

Both tables and figures clearly show that the non-uniformity parameter has a remarkable influence on the flexural–torsional buckling load. For each selected power-law exponent, nonlocal parameter and loading position, the stability values of prismatic beams with α=β=0 and of double-tapered ones with α=β=1 are the lowest and highest, respectively. By consulting Table 3, Table 4 and Table 5 and Figure 6 and Figure 7, one can also note that the response is much more affected by the flange tapering parameter β than the web non-uniformity ratio α, since the lowest flexural–torsional buckling modes usually occur with the lowest axis moment of inertia.

For beam-columns subjected to an axial load on the centroid and on the TF for x=0, it is found that the buckling parameter increases with increasing values of both α and β, due to the enhancement of the cross-section’s geometrical properties along with an overall increase in flexural and torsional stiffness in the elastic member (see Table 3 and Table 4). The sensitivity of the mechanical response is slightly different for I-beams with an axial load applied on the TF at x=L. As shown in Table 5 and Figure 7, the linear stability strength of beams with a constant flange width tends to reduce with increasing values of α. This is mainly related to the fact that the initial bending moment $M_{y}^{*}$ due to axial load eccentricity is enhanced by increasing the web tapering ratio α. The effect of this phenomenon on the buckling resistance of web and flange tapered beams is quite negligible when all the section walls have the same non-uniformity ratio (i.e., for α=β).

As was also expected, the nonlocal parameter shows a stiffness-softening effect and reduces the buckling strength for all the selected loading positions. Based on the plots in Figure 3, Figure 4, Figure 5, Figure 6 and Figure 7, it seems that the effect of the Eringen’s nonlocal parameter on the buckling response is more pronounced at higher tapering ratios and gradient indexes, especially for beams made of pure ceramic. For example, the normalized buckling load of prismatic members in Table 5 with m=1.2 decreases by 36.5% when μ increases from 0 to 3. This can be explained by the fact that the flexural and torsional stiffness of simply supported tapered I-beams in a nonlocal theoretical context is inversely proportional to the Eringen’s parameter. Usually, the introduction of a nonlocal effect increases the deflection response, or this increase is equivalent to the stiffness reduction in the structural member. Since the linear buckling resistance of beams is directly proportional to their stiffness, a meaningful decrease in the critical load is expected. In Figure 11, we represent the effect of the nonlocality parameter on the first four flexural–torsional buckling loads of nonlocal thin-walled beams with an I-section. In this way, we account for a compressive axial load applied at the TF for x=0, while considering both an AFG prismatic beam with m=1 and a homogeneous tapered beam with α=β=0.5. Based on the plots in Figure 11, it is clear that the nonlocal parameter has more pronounced effects on higher flexural–torsional buckling modes when compared with the lowest ones. It can also be stated that it is necessary to rely on nonlocal theories for an accurate estimation of the flexural–torsional stability limit of nanosized thin-walled beams at higher buckling modes. In addition, it is clearly observable that the nonlocal parameter effect increases when the μ ranges between 0 and 0.9.

## 5. Conclusions

In this paper, we explore the flexural–torsional buckling of AFG nanobeams with a varying I-section by resorting to the Vlasov model and Eringen’s nonlocal elasticity theory. The material properties vary in the axial direction of structural elements according to the power-law distribution of the material constituents. The principle of minimum potential energy is applied to determine the governing equilibrium equations and boundary conditions for AFG tapered thin-walled nanobeams subjected to eccentric axial loads. The governing equations of the problem are implemented and solved numerically by means of the DQM in order to determine the flexural–torsional buckling load. A broad systematic investigation checks for the influence of some important parameters, including the web and flange tapering ratios, the nonlocal parameter, the mode number, the axial load eccentricity and the non-homogeneity index, on the overall response of doubly symmetric tapered nanobeams subjected to simply supported boundary conditions. For all the selected loading positions, it is found that the flexural–torsional buckling capacity of nanobeams with a tapered I-section decreases as the nonlocal parameter increases, whereas the buckling load increases as the flange tapering ratio and ceramic phase, Al_2_O_3_, increase. The effect of the flange tapering parameter *β* on the buckling capacity is more pronounced than that related to the web tapering ratio, *α*. As expected, the buckling capacity reaches its highest value in the absence of all possible eccentricity. In addition, the elastic buckling capacity of homogeneous double-tapered beams decreases as the nonlocal parameter increases, especially when compared to AFG prismatic I-beams. The small-scale effects become even more pronounced at higher buckling modes, such that they cannot be clearly disregarded when accurately defining the problem. In its present state, the formulation does not consider the grain sizes or the shapes of the alumina or aluminum components, but this will be considered in a more extended formulation that will include possible material anisotropies. A further extension of the proposed formulation will include the nonlinear effects on the coupled thermo-mechanical stability of tapered micro/nanosized systems, accounting for the possible presence of porosities and defects, together with different boundary and environmental conditions.

## Figures and Tables

**Figure 1 nanomaterials-11-01936-f001:**
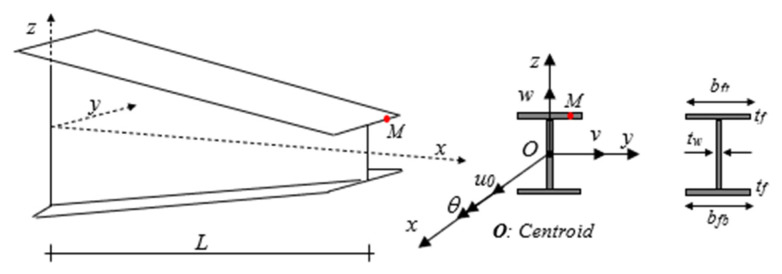
Geometrical scheme of variable doubly symmetric I-section beam—coordinate system and notation for the displacement parameters.

**Figure 2 nanomaterials-11-01936-f002:**
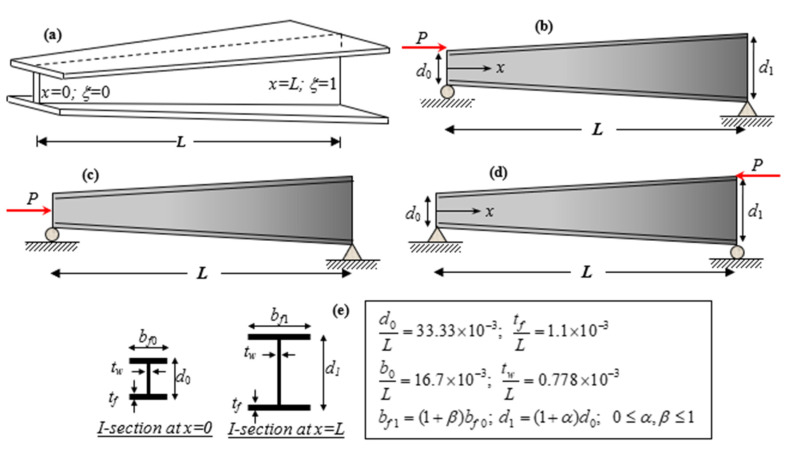
(**a**) Schematic representation of double-tapered beam with I-sections. (**b**) Axial load on the TF at *x* = 0 (**c**) Axial load on the centroid. (**d**) Axial load on the TF at *x* = *L* (**e**) Geometrical properties.

**Figure 3 nanomaterials-11-01936-f003:**
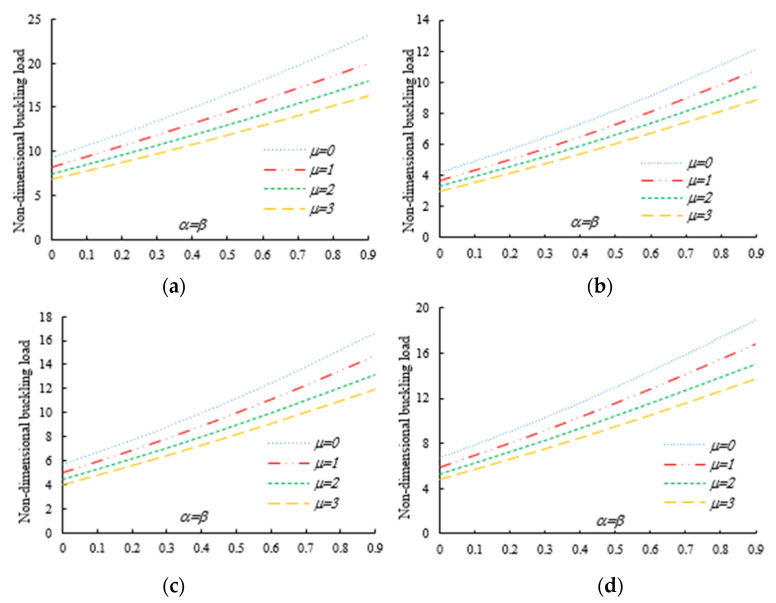
Variation in the flexural–torsional buckling load (*P_nor_*) of I-tapered nanobeams with varying tapering and nonlocality parameters—Figure 2. Effect of the axial load eccentricity, material graduation and tapering parameter on the normalized buckling load *P_nor_* for simply supported thin-walled nanobeams subjected to a constant compressive load with two different nonlocal parameters: (**a**)Homogeneous (Alumina); (**b**) m=0.6; (**c**) m=1.3; (**d**) m=2.

**Figure 4 nanomaterials-11-01936-f004:**
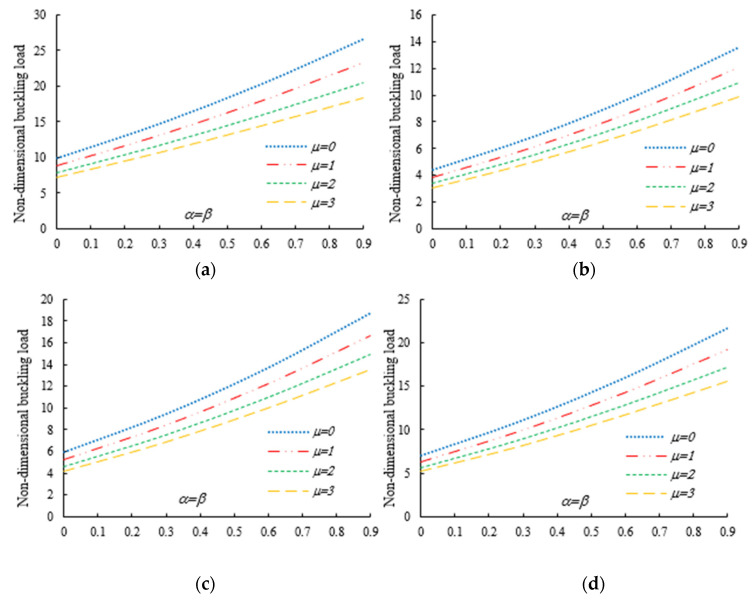
Variation in the flexural buckling load *P_nor_* of I-tapered nanobeams with variations in the tapering and nonlocality parameters for different material indexes (axial load on the centroid) (**a**) Homogeneous (Alumina); (**b**) m=0.6; (**c**) m=1.3; (**d**) m=2.

**Figure 5 nanomaterials-11-01936-f005:**
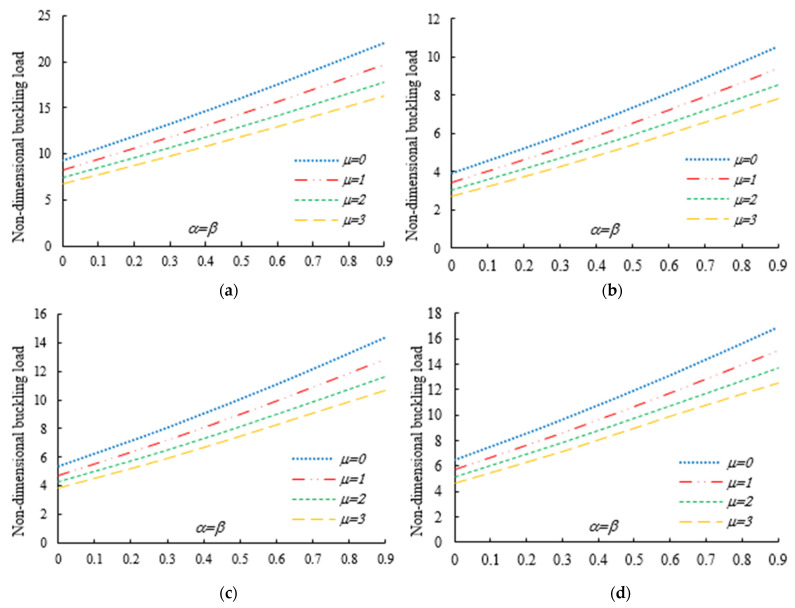
Variation in the flexural–torsional buckling load *P_nor_* of I-tapered nanobeams with variations in the tapering and nonlocality parameters for different material indexes (axial load on the TF at *x* = *L*): (**a**) Homogeneous (Alumina); (**b**) m=0.6; (**c**) m=1.3; (**d**) m=2.

**Figure 6 nanomaterials-11-01936-f006:**
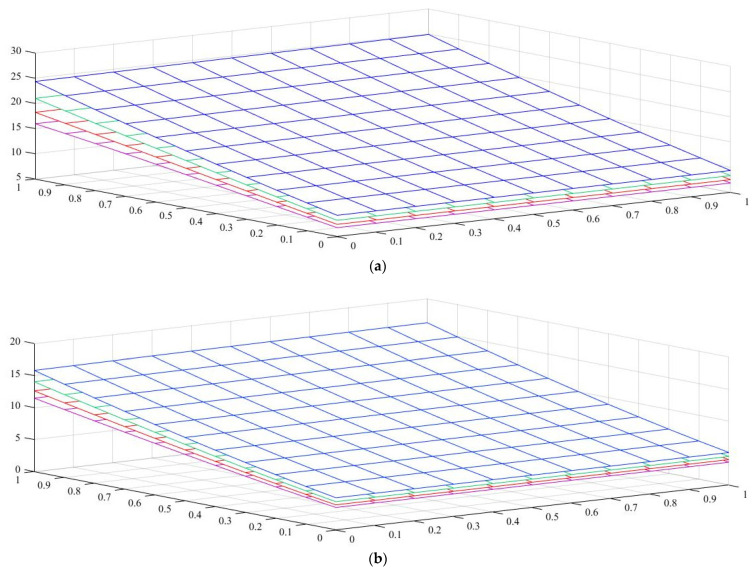
Variation in normalized buckling load $\left(P_{nor}\right)$ of local and nonlocal beams with a tapered I-section for various webs. Effect of the power-law exponent and tapering parameter on the normalized torsional buckling load $\left(P_{nor}\right)$ of simply supported thin-walled nanobeams with different nonlocal parameters (axial load applied on the Centroid): (**a**) pure ceramic, (**b**) AFG $\left(m=1\right)$.

**Figure 7 nanomaterials-11-01936-f007:**
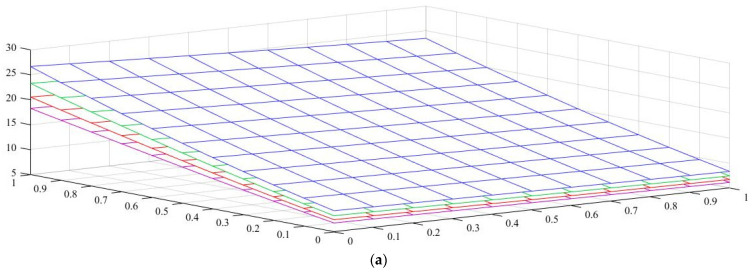
Variation in the normalized buckling load $\left(P_{nor}\right)$ of local and nonlocal beams with a tapered I-section for various web and flange tapering ratios and four different Eringen’s parameters (axial load on the TF at *x* = *L*): (**a**) pure ceramic, (**b**) AFG $\left(m=1\right)$.

**Figure 8 nanomaterials-11-01936-f008:**
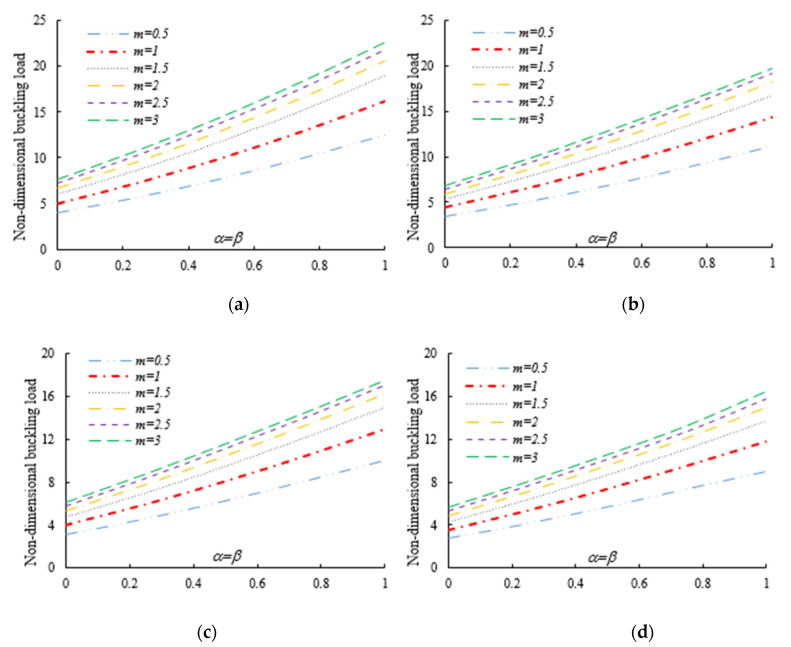
Variation in the flexural–torsional buckling load of I-tapered nanobeams with the tapering ratio and power-law exponent, for different nonlocality parameters (axial load on the TF at x=0): (**a**) μ=0; (**b**) μ=1; (**c**) μ=2; (**d**) μ=3.

**Figure 9 nanomaterials-11-01936-f009:**
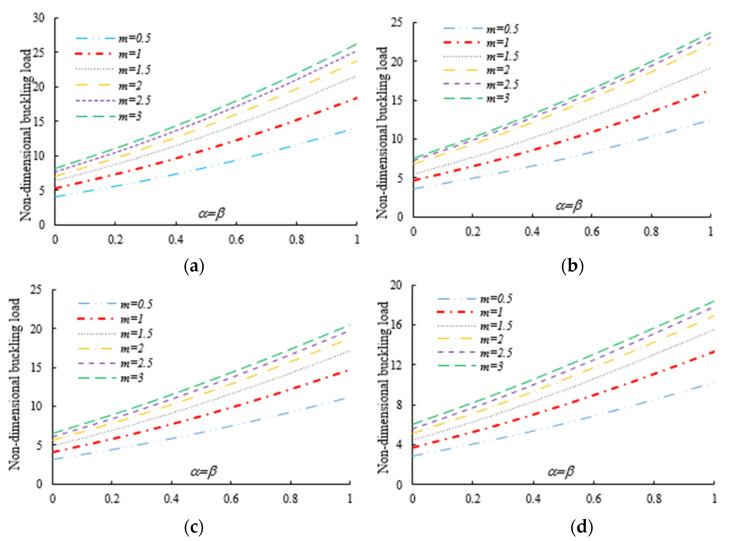
Variation in the flexural–torsional buckling load of I-tapered nanobeams with the tapering ratio and power-law exponent, for different nonlocality parameters (axial load on the centroid): (**a**) μ=0; (**b**) μ=1; (**c**) μ=2; (**d**) μ=3.

**Figure 10 nanomaterials-11-01936-f010:**
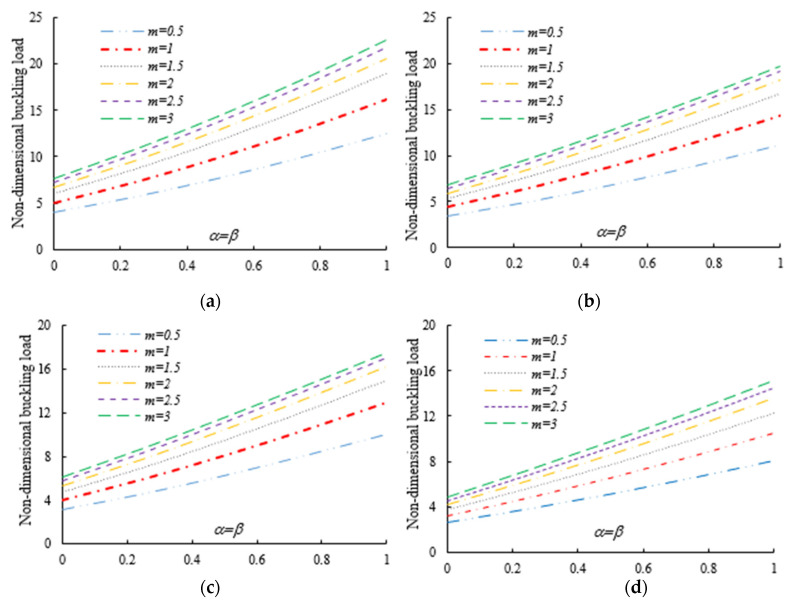
Variation in the flexural–torsional buckling load of I-tapered nanobeams with the tapering ratio and power-law exponent, for different nonlocality parameters (axial load on the TF at *x* = *L*): (**a**) μ=0; (**b**) μ=1; (**c**) μ=2; (**d**) μ=3.

**Figure 11 nanomaterials-11-01936-f011:**
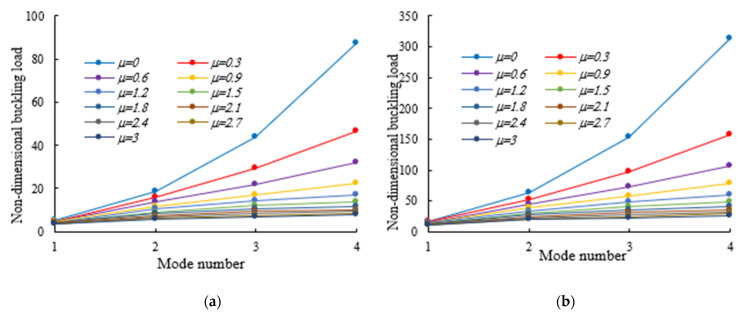
Effects of Eringen’s nonlocal parameters on the flexural–torsional buckling load of doubly symmetric I-section nanobeams under a compressive axial load on the TF at x = 0: (**a**) α=β=0; m=1; (**b**) α=β=0.5; Homogeneous.

**Table 1 nanomaterials-11-01936-t001:** Dimensionless buckling load $\left(P_{nor}\right)$ for local tapered homogenous I-beams (alumina) with different tapering parameters and loading positions.

Axial Load Position	*α* = *β*	DQM					ANSYS [104]
		Number of Points along x-Direction					
		*n* = 5	*n* = 10	*n* = 15	*n* = 20	*n* = 30	
Centroid	0	9.824	9.870	9.870	9.870	9.870	9.866
	0.2	12.970	13.006	13.006	13.006	13.006	12.997
	0.4	16.550	16.494	16.494	16.494	16.494	16.466
	0.6	20.647	20.326	20.326	20.326	20.326	20.276
	0.8	25.426	24.497	24.497	24.497	24.497	24.414
	1.0	31.184	29.004	29.003	29.003	29.003	27.605
TF of left end section	0	9.213	9.248	9.248	9.248	9.248	9.274
	0.2	11.974	11.964	11.964	11.964	11.964	12.000
	0.4	15.050	14.895	14.895	14.895	14.895	14.907
	0.6	18.570	18.029	18.029	18.029	18.029	18.050
	0.8	22.915	21.358	21.357	21.357	21.357	21.399
	1.0	29.257	24.877	24.876	24.876	24.876	24.956
TF of right end section	0	9.213	9.248	9.248	9.248	9.248	9.274
	0.2	11.805	11.862	11.862	11.862	11.862	11.922
	0.4	14.557	14.610	14.610	14.610	14.610	14.690
	0.6	17.524	17.469	17.469	17.469	17.469	17.638
	0.8	20.912	20.429	20.429	20.429	20.429	20.737
	1.0	25.283	23.488	23.489	23.489	23.489	23.992

**Table 2 nanomaterials-11-01936-t002:** Effect of the axial load eccentricity, material graduation and tapering parameter on the normalized buckling load $\left(P_{nor}\right)$ for simply supported thin-walled nanobeams subjected to a constant compressive load with two different nonlocal parameters.

*μ* (nm^2^)	*α* = *β*	Axial Load on the TF at x=0				Axial Load on the Centroid				Axial Load on the TF at x=L			
		Homogeneous	*m* = 0.8	*m* = 1.6	*m* = 2.4	Homogeneous	*m* = 0.8	*m* = 1.6	*m* = 2.4	Homogeneous	*m* = 0.8	*m* = 1.6	*m* = 2.4
0	0.0	9.248	4.602	6.165	7.151	9.870	4.816	6.489	7.571	9.248	4.383	5.863	6.843
	0.3	13.416	7.140	9.516	10.879	14.711	7.641	10.272	11.833	13.232	6.596	8.804	10.165
	0.6	18.070	10.143	13.401	15.154	20.338	11.104	14.836	16.929	17.506	9.086	12.071	13.839
	0.9	23.186	13.578	17.787	19.927	26.738	15.194	20.169	22.823	22.029	11.827	15.648	17.827
2.0	0.0	7.453	3.595	4.836	5.684	7.936	3.756	5.088	6.028	7.453	3.389	4.714	5.705
	0.3	10.646	5.744	7.658	8.754	11.750	6.102	8.204	9.471	10.679	5.180	6.899	8.010
	0.6	14.010	8.171	10.752	12.080	15.996	8.922	11.902	13.547	14.110	7.261	9.655	11.113
	0.9	17.488	10.850	14.071	15.572	20.591	12.178	16.080	18.061	17.713	9.558	12.662	14.450

**Table 3 nanomaterials-11-01936-t003:** Effect of the power-law exponent and tapering parameter on the normalized flexural–torsional buckling load $\left(P_{nor}\right)$ of simply supported thin-walled nanobeams with different nonlocal parameters (axial load applied on the TF at x=0).

*μ* (nm^2^)	*α*	*m* = 0.6				*m* = 1.2				*m* = 1.8			
		*β* = 0	*β* = 0.2	*β* = 0.5	*β* = 0.8	*β* = 0	*β* = 0.2	*β* = 0.5	*β* = 0.8	*β* = 0	*β* = 0.2	*β* = 0.5	*β* = 0.8
0	0.0	4.057	5.488	7.943	10.725	5.483	7.413	10.693	14.373	6.462	8.685	12.418	16.559
	0.2	4.065	5.503	7.973	10.778	5.496	7.435	10.739	14.454	6.479	8.713	12.475	16.658
	0.5	4.077	5.524	8.015	10.852	5.514	7.467	10.802	14.565	6.502	8.754	12.554	16.793
	0.8	4.089	5.543	8.053	10.919	5.531	7.496	10.859	14.664	6.524	8.790	12.625	16.914
1.0	0.0	3.588	4.886	7.089	9.543	4.842	6.598	9.539	12.765	5.715	7.735	11.062	14.650
	0.2	3.595	4.899	7.115	9.590	4.852	6.617	9.578	12.835	5.729	7.760	11.111	14.736
	0.5	3.605	4.916	7.151	9.654	4.868	6.643	9.632	12.932	5.748	7.793	11.179	14.854
	0.8	3.614	4.933	7.184	9.713	4.881	6.668	9.682	13.020	5.765	7.824	11.242	14.962
3.0	0.0	2.899	4.008	5.853	7.848	3.886	5.405	7.871	10.461	4.586	6.346	9.109	11.916
	0.2	2.904	4.017	5.873	7.885	3.893	5.418	7.901	10.516	4.594	6.363	9.147	11.985
	0.5	2.911	4.030	5.901	7.936	3.902	5.437	7.943	10.594	4.605	6.387	9.200	12.083
	0.8	2.917	4.042	5.927	7.985	3.910	5.454	7.982	10.667	4.615	6.409	9.250	12.173

**Table 4 nanomaterials-11-01936-t004:** Effect of the power-law exponent and tapering parameter on the normalized flexural–torsional buckling load $\left(P_{nor}\right)$ of simply supported thin-walled nanobeams with different nonlocal parameters (axial load applied on the centroid).

*μ* (nm^2^)	*α*	*m* = 0.6				*m* = 1.2				*m* = 1.8			
		*β* = 0	*β* = 0.2	*β* = 0.5	*β* = 0.8	*β* = 0	*β* = 0.2	*β* = 0.5	*β* = 0.8	*β* = 0	*β* = 0.2	*β* = 0.5	*β* = 0.8
0	0.0	4.243	5.837	8.689	12.096	5.756	7.928	11.796	16.392	6.815	9.345	13.814	19.077
	0.2	4.244	5.838	8.690	12.097	5.757	7.929	11.797	16.392	6.816	9.346	13.815	19.078
	0.5	4.245	5.839	8.691	12.098	5.759	7.930	11.798	16.394	6.817	9.348	13.817	19.079
	0.8	4.246	5.840	8.692	12.099	5.760	7.931	11.800	16.395	6.819	9.349	13.818	19.081
1.0	0.0	3.736	5.177	7.741	10.775	5.054	7.023	10.504	14.589	5.989	8.287	12.299	16.949
	0.2	3.736	5.178	7.742	10.776	5.055	7.024	10.505	14.589	5.990	8.288	12.300	16.950
	0.5	3.737	5.179	7.743	10.777	5.056	7.026	10.506	14.590	5.992	8.289	12.302	16.951
	0.8	3.738	5.180	7.744	10.777	5.058	7.027	10.507	14.592	5.993	8.291	12.303	16.953
3.0	0.0	2.990	4.210	6.353	8.840	4.000	5.688	8.610	11.945	4.725	6.716	10.077	13.823
	0.2	2.990	4.210	6.353	8.840	4.001	5.689	8.611	11.946	4.726	6.717	10.078	13.823
	0.5	2.991	4.211	6.354	8.841	4.002	5.690	8.612	11.947	4.727	6.718	10.079	13.824
	0.8	2.992	4.212	6.355	8.842	4.003	5.691	8.613	11.948	4.729	6.719	10.080	13.826

**Table 5 nanomaterials-11-01936-t005:** Effect of the power-law exponent and tapering parameter on the normalized flexural–torsional buckling load $\left(P_{nor}\right)$ of simply supported thin-walled nanobeams with different nonlocal parameters (axial load applied on the TF at x=l).

*μ* (nm^2^)	*α*	*m* = 0.6				*m* = 1.2				*m* = 1.8			
		*β* = 0	*β* = 0.2	*β* = 0.5	*β* = 0.8	*β* = 0	*β* = 0.2	*β* = 0.5	*β* = 0.8	*β* = 0	*β* = 0.2	*β* = 0.5	*β* = 0.8
0	0.0	3.881	5.299	7.808	10.770	5.218	7.133	10.513	14.484	6.160	8.392	12.304	16.869
	0.2	3.785	5.158	7.585	10.447	5.080	6.934	10.200	14.037	5.995	8.156	11.939	16.352
	0.5	3.648	4.961	7.277	10.005	4.889	6.658	9.775	13.434	5.766	7.830	11.441	15.652
	0.8	3.526	4.785	7.005	9.621	4.719	6.416	9.404	12.912	5.564	7.544	11.007	15.046
1.0	0.0	3.378	4.661	6.928	9.595	4.509	6.250	9.312	12.894	5.320	7.358	10.909	15.020
	0.2	3.288	4.531	6.726	9.308	4.382	6.066	9.030	12.498	5.168	7.141	10.581	14.566
	0.5	3.164	4.352	6.450	8.917	4.209	5.819	8.650	11.965	4.963	6.849	10.138	13.953
	0.8	3.054	4.195	6.209	8.577	4.060	5.605	8.323	11.507	4.787	6.598	9.756	13.423
3.0	0.0	2.659	3.748	5.668	7.908	3.465	4.968	7.585	10.607	4.038	5.832	8.891	12.354
	0.2	2.584	3.640	5.501	7.678	3.365	4.818	7.354	10.292	3.924	5.656	8.624	12.000
	0.5	2.484	3.494	5.276	7.364	3.235	4.621	7.047	9.868	3.779	5.429	8.270	11.519
	0.8	2.400	3.370	5.082	7.093	3.128	4.457	6.787	9.505	3.658	5.239	7.968	11.103

**Table 6 nanomaterials-11-01936-t006:** Effect of the power-law exponent and tapering parameter on the normalized torsional buckling load $\left(P_{nor}\right)$ of simply supported thin-walled nanobeams with different nonlocal parameters (axial load applied on the Centroid).

	*α*	*μ* = 0					*μ* = 0.5					*μ* = 1.0				
		*β* = 0	*β* = 0.25	*β* = 0.5	*β* = 0.75	*β* = 1.0	*β* = 0	*β* = 0.25	*β* = 0.5	*β* = 0.75	*β* = 1.0	*β* = 0	*β* = 0.25	*β* = 0.5	*β* = 0.75	*β* = 1.0
*m* = 1	0.0	26.399	31.400	36.266	41.068	45.851	23.116	28.587	33.610	38.395	43.059	19.151	25.731	31.080	35.936	40.536
	0.25	23.264	28.179	33.116	38.108	43.174	20.578	25.743	30.737	35.666	40.588	17.772	23.395	28.526	33.445	38.266
	0.5	21.048	25.814	30.706	35.739	40.919	18.770	23.667	28.555	33.493	38.514	16.526	21.651	26.582	31.464	36.362
	0.75	19.410	24.025	28.831	33.836	39.040	17.424	22.099	26.863	31.755	36.793	15.526	20.319	25.074	29.885	34.787
	1.0	18.155	22.633	27.343	32.291	37.476	16.386	20.881	25.526	30.351	35.365	14.726	19.276	23.882	28.610	33.484
*m* = 2	0.0	32.917	39.157	45.200	51.149	57.068	28.810	35.822	42.067	47.952	53.660	22.425	32.135	39.040	45.017	50.590
	0.25	29.212	35.391	41.552	47.747	54.009	25.880	32.480	38.722	44.811	50.846	21.761	29.544	36.068	42.145	48.015
	0.5	26.565	32.587	38.713	44.971	51.374	23.745	30.005	36.136	42.258	48.427	20.660	27.504	33.757	39.811	45.799
	0.75	24.596	30.445	36.477	42.709	49.145	22.138	28.118	34.109	40.187	46.387	19.612	25.915	31.943	37.923	43.935
	1.0	23.080	28.767	34.688	40.853	47.264	20.890	26.642	32.492	38.494	44.671	18.718	24.658	30.496	36.382	42.371
*m* = 3	0.0	36.991	43.765	50.277	56.664	63.005	32.637	40.321	46.990	53.227	59.259	24.922	36.356	43.842	50.097	55.888
	0.25	32.952	39.692	46.349	53.002	59.699	29.420	36.657	43.364	49.843	56.229	24.530	33.542	40.580	46.993	53.131
	0.5	30.034	36.619	43.249	49.973	56.819	27.038	33.908	40.519	47.052	53.592	23.509	31.257	38.008	44.432	50.724
	0.75	27.845	34.249	40.783	47.482	54.361	25.229	31.794	38.266	44.764	51.346	22.406	29.456	35.971	42.336	48.675
	1.0	26.151	32.379	38.794	45.421	52.272	23.816	30.130	36.454	42.879	49.441	21.420	28.021	34.335	40.613	46.942
Homogeneous	0.0	55.172	61.392	67.658	73.998	80.419	52.003	57.786	63.516	69.265	75.059	49.179	54.536	59.728	64.886	70.053
	0.25	49.161	55.583	62.099	68.739	75.511	46.296	52.418	58.496	64.612	70.798	43.724	49.607	55.278	60.896	66.524
	0.5	44.446	50.894	57.492	64.265	71.217	41.822	48.036	54.265	60.575	66.994	39.462	45.506	51.410	57.301	63.234
	0.75	40.765	47.145	53.724	60.521	67.539	38.353	44.530	50.784	57.168	63.700	36.189	42.219	48.196	54.216	60.321
	1.0	37.852	44.126	50.635	57.395	64.408	35.627	41.713	47.927	54.311	60.880	33.635	39.582	45.549	51.611	57.798

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
