# Peer review of "Nonlocal Analysis of the Flexural–Torsional Stability for FG Tapered Thin-Walled Beam-Columns"

_nanomaterials, 2021, doi:10.3390/nano11081936_

Round 1
Reviewer 1 Report
The authors present a study regarding the flexural-torsional stability of functionally graded thin walled beams. The approach is based on the Vlasov theory, which takes into account the coupling between the axial and bending stress. The equilibrium equations are solved numerically using DMQ. This is used to determine the flexural-torsional buckling load. Several parameters are varied, such as axial load eccentricity, mode number and tapering ratio and the response of the thin-walled FG nanobeams is monitored.
The paper presents a rather technical investigation regarding the aforementioned topics. In this respect I have several comment:
1. The paper is abundant in data, which makes it a little hard to follow. I would suggest to reorganizing Section 4, potentially be including part of the information into an Appendix or Supplementary material. I would urge the authors to focus on the differences regarding the outcome of changing the parameters, rather than reproducing the corresponding data for each of them.
2. The authors should also comment on the limitations of the present model, particularly with respect to the finite grain sizes, typically found in the considered materials (Al2O3 and Al). Here, the authors use Young moduli as material constants for each material. Typical system sizes should be indicated.
3. Can anisotropic effects be included in the current approach ? E.g. porous alumina may exhibit anisotropic effects.
Author Response
We thank the Reviewers for their valuable comments, which helped the authors to improve the quality of the manuscript. The detailed responses are reported here below
Reviewer #1
The authors present a study regarding the flexural-torsional stability of functionally graded thin walled beams. The approach is based on the Vlasov theory, which takes into account the coupling between the axial and bending stress. The equilibrium equations are solved numerically using DMQ. This is used to determine the flexural-torsional buckling load. Several parameters are varied, such as axial load eccentricity, mode number and tapering ratio and the response of the thin-walled FG nanobeams is monitored. The paper presents a rather technical investigation regarding the aforementioned topics. In this respect I have several comment:
Comment: The paper is abundant in data, which makes it a little hard to follow. I would suggest to reorganizing Section 4, potentially be including part of the information into an Appendix or Supplementary material. I would urge the authors to focus on the differences regarding the outcome of changing the parameters, rather than reproducing the corresponding data for each of them.
Response: The authors thank the reviewer for his suggestion, but the numerical investigation in Section 4 represents the core aspect of the work, with a preliminary validation of the proposed nonlocal model compared to conventional finite elements and predictions from literature, and systematic analysis aimed at checking the sensitivity of the equivalent flexural-torsional response of the structure for different input parameters. Both parts of the numerical investigation are equally important, and the authors would prefer to leave the organization of the work unaltered.
Comment: The authors should also comment on the limitations of the present model, particularly with respect to the finite grain sizes, typically found in the considered materials (Al2O3 and Al). Here, the authors use Young moduli as material constants for each material. Typical system sizes should be indicated.
Response: The authors thank the reviewer for his suggestion. At the present state, the formulation does not consider the exact grain size and shape within Al2O3 ceramics or Al materials, but considers some equivalent mechanical properties from literature. In a further work we will focus on this aspect within a more extended formulation which correlates the selected size-dependent numerical parameters with the physical micro/nanostructural nature of the material and its mechanical properties, even including possible porosities and defects, despite the expected increase of the computational effort. An additional comment about such limitation has been introduced in the revised version of the work.
Comment: Can anisotropic effects be included in the current approach ? E.g. porous alumina may exhibit anisotropic effects.
Response: At the present state, the approach does not consider possible anisotropic effects, but a numerical homogenisation method can be applied to establish a mutual correlation between the microstructural material pattern and the anisotropy stiffness properties. This aspect will be considered by the authors in the next work, possibly supported by experimental predictions.
Reviewer 2 Report
This paper is focused onto the flexural-torsional stability of functionally graded (FG) nonlocal thin-walled beam-columns with a tapered I-section. Therewith, the paper addresses several up-to-date topics from the field of structural mechanics. The equilibrium equations are solved numerically by means of the differential quadrature method (DQM). The critical buckling loads are determined. Also, a parameter study has been done to check the influence of different parameters.
The paper is well written and structured. It would be definitely of interest to the research community. I only have a few minor suggestions to the authors:
- English, although generally good, could be polished in a few places. For instance, line 268: "...acting at end beam..." should probably be: "...acting at the end of the beam...", or at the beam end, etc.
- At the end of the introduction, the sentence: "This paper can be considered as extension of the stability study in Ref. [100] for non-prismatic thin-walled nanobeam-columns with an arbitrary distribution of the material properties in the axial direction, whose numerical outcomes could be useful for future developments on thin-walled structures" should be the beginning of a new paragraph. In that paragraph the novelty of the paper could be emphasized a bit more.
- The authors could check these references: https://doi.org/10.22190/FUME190415008P, https://doi.org/10.1515/cls-2019-0018
- Lines 254-255: "...in the context of small displacements...". But later, in lines 261-262, the Green strain tenson, with quadratic strain terms, is introduced and further used. If it is dealt with small displacements, why is it needed to take the quadratic strain terms typically used in nonlinear analysis.
- The term u0 in Eq. (4a) is not explained (I assume the subscripting zero is a typo).
- Some outlook for further work could be added in conclusions.
Author Response
We thank the Reviewers for their valuable comments, which helped the authors to improve the quality of the manuscript. The detailed responses are reported here below
Reviewer #2
This paper is focused onto the flexural-torsional stability of functionally graded (FG) nonlocal thin-walled beam-columns with a tapered I-section. Therewith, the paper addresses several up-to-date topics from the field of structural mechanics. The equilibrium equations are solved numerically by means of the differential quadrature method (DQM). The critical buckling loads are determined. Also, a parameter study has been done to check the influence of different parameters. The paper is well written and structured. It would be definitely of interest to the research community. I only have a few minor suggestions to the authors:
Comment #1: English, although generally good, could be polished in a few places. For instance, line 268: "...acting at end beam..." should probably be: "...acting at the end of the beam...", or at the beam end, etc.
Response #1: The authors agree with the reviewers and have properly checked and polished the English language and grammar in the text.
Comment #2: At the end of the introduction, the sentence: "This paper can be considered as extension of the stability study in Ref. [100] for non-prismatic thin-walled nanobeam-columns with an arbitrary distribution of the material properties in the axial direction, whose numerical outcomes could be useful for future developments on thin-walled structures" should be the beginning of a new paragraph. In that paragraph the novelty of the paper could be emphasized a bit more.
Response #2: The authors agree with the reviewer and have postponed this sentence in the next section, while stressing the novel aspect of the paper.
Comment #3: The authors could check these references: https://doi.org/10.22190/FUME190415008P, https://doi.org/10.1515/cls-2019-0018
Response #3: Both references have been now considered in the revised version of the work for affinity reasons.
Comment #4: Lines 254-255: "...in the context of small displacements...". But later, in lines 261-262, the Green strain tenson, with quadratic strain terms, is introduced and further used. If it is dealt with small displacements, why is it needed to take the quadratic strain terms typically used in nonlinear analysis.
Response #4: The formulation is here written in a more generalized version, accounting for the non-linear strain field to define the strain energy due to the initial stresses, in a variational form. It is clear that the nonlinear terms must be disregarded in the equilibrium equations for a linear stability study, as highlighted in the text, whereas they are strictly required by nonlinear analyses.
Comment #5: The term u0 in Eq. (4a) is not explained (I assume the subscripting zero is a typo).
Response #5: The authors confirm that the subscripting zero was a typo.
Comment #6: Some outlook for further work could be added in conclusions.
Response #6: The authors thank the reviewer for his suggestion, and have pointed out the possible extension of the proposed formulation at the end of the concluding section.
Round 2
Reviewer 1 Report
The authors responded to my comments and included, in part, the suggestions that I mentioned as possible future extentions. The original structure of the paper was kept, but I leave this to the authors to decide. This can also be an option. Overall the paper has improved and can now be accepted for publication.